# Metamorphosis of memory circuits in *Drosophila* reveals a strategy for evolving a larval brain

**James W Truman[1,2]\*, Jacquelyn Price[1], Rosa L Miyares[1], Tzumin Lee[1,3]**

[1]Janelia Research Campus, Ashburn, United States; [2]Department of Biology, Friday Harbor Laboratories, University of Washington, Friday Harbor, United States; [3]Life Sciences Institute, University of Michigan, Ann Arbor, United States

**Abstract** Mushroom bodies (MB) of adult *Drosophila* have a core of thousands of Kenyon neurons; axons of the early-born g class form a medial lobe and those from later-born α'β' and αβ classes form both medial and vertical lobes. The larva, however, hatches with only γ neurons and forms a vertical lobe 'facsimile' using larval-specific axon branches from its γ neurons. MB input (MBINs) and output (MBONs) neurons divide the Kenyon neuron lobes into discrete computational compartments. The larva has 10 such compartments while the adult has 16. We determined the fates of 28 of the 32 MBONs and MBINs that define the 10 larval compartments. Seven compartments are subsequently incorporated into the adult MB; four of their MBINs die, while 12 MBINs/ MBONs remodel to function in adult compartments. The remaining three compartments are larval specific. At metamorphosis their MBIN/MBONs trans-differentiate, leaving the MB for other adult brain circuits. The adult vertical lobes are made *de novo* using MBONs/MBINs recruited from pools of adult-specific neurons. The combination of cell death, compartment shifting, trans-differentiation, and recruitment of new neurons result in no larval MBIN-MBON connections being maintained through metamorphosis. At this simple level, then, we find no anatomical substrate for a memory trace persisting from larva to adult. The adult phenotype of the trans-differentiating neurons represents their evolutionarily ancestral phenotype while their larval phenotype is a derived adaptation for the larval stage. These cells arise primarily within lineages that also produce permanent MBINs and MBONs, suggesting that larval specifying factors may allow information related to birth-order or sibling identity to be interpreted in a modified manner in the larva to allow these neurons to acquire larval phenotypic modifications. The loss of such factors at metamorphosis then allows these neurons to revert to their ancestral functions in the adult.

**\*For correspondence:**
jwt@uw.edu

**Competing interest:** The authors declare that no competing interests exist.

## Editor's evaluation

The complete metamorphosis of higher insects is one of the most fascinating and complex processes in nature. The discrepancy in form and function between larvae, pupa, and adult insects is breathtaking, begging the question of how these forms and functions can so seamlessly follow each other. For the highest-order brain center of the insects, the mushroom body, the authors provide a masterpiece analysis of this process at the cellular level. Given the breadth and depth of the data that the authors present, the current study will serve as a reference for the field of developmental neuroscience for many years to come; indeed, the study data are eagerly awaited in the field.

**Figure 1.** Comparison of the effects of direct development versus metamorphosis on neurogenesis and the establishment of neuronal phenotypes. In a direct developing insect like a cricket, embryogenesis produces a hatchling with a miniature cricket body plan. Its neuroblasts (NBs) generate their entire lineages during embryogenesis so that at hatching the CNS has its full complement of neurons and they already possess their mature phenotypes. In the metamorphic development of *Drosophila*, by contrast, a shortened embryonic phase redirects development to produce a simplified, larval body plan. Their neuroblasts produce only their early-born neuron types and neuronal phenotypes are modified for larval morphology and behavior. During larval growth, the arrested NBs reactivate to produce the rest of their neuronal lineages, but their young neurons arrest development soon after their birth. The species-typical body plan of the fly finally arises at metamorphosis and, in the CNS, remodeling larval neurons and maturing postembryonic-born neurons combine to make the mature nervous system of the fly.

## Introduction

Direct developing insects like crickets and grasshoppers produce a species-typical body form during embryogenesis and hatch as a miniature version of the adult but lack wings and genitalia. Holometabolous insects, which have complete metamorphosis, have modified their embryogenesis to produce a simplified larval body instead (*Figure 1*). In the larval body, embryonic fields, which had been fully utilized to generate their adult-like structures, are only partially patterned, with the patterned portion making the larval structure and the remainder preserved through larval growth to become imaginal discs or primordia. The imaginal primordia, along with some larval cells, combine to construct the species-typical body form at metamorphosis.

Direct development also results in a hatchling possessing an essentially mature, but miniature brain, which is used for both the nymphal and adult stages (reviews by *Kutsch and Heckmann, 1995*; *Truman, 2005*). The evolution of the larva, though, altered brain development to make a modified, simpler brain appropriate for the sensory and motor demands of the larva. This larval brain, though, is not discarded at the end of larval growth and a new one made from scratch. Most larval neurons persist and some, like the interneurons mediating backwards locomotion, have similar functions in both larva and adult (*Lee and Doe, 2021*), but, as we show in this paper, the maintenance of neuronal function from larva to adult is not always the case. At metamorphosis, recycled larval cells are combined with adult-specific neurons to make the nervous system of the adult (*Truman, 2005*).

Although the focus on nervous system metamorphosis is usually on the postembryonic transformation of the larval brain into that of the adult, complementary changes had to have occurred during embryogenesis to generate the modified larval brain. One key embryonic change involved neurogenesis (*Figure 1*). The central brain and ventral nerve cord (VNC) of insects arise from a fixed number of neuronal stem cells (neuroblasts [NBs]), with about 100 per brain hemisphere (*Urbach and Technau, 2003*) and about 30 per segmental hemineuropil (*Thomas et al., 1984*). Each NB makes a characteristic set of neurons in a defined temporal order (*Kohwi and Doe, 2013*; *Doe, 2017*; *Miyares and Lee, 2019*; *Rossi et al., 2021*). The sets of NBs in the brain and VNC, though, are highly conserved and were established well before the evolution of the larva and complete metamorphosis (*Thomas et al., 1984*; *Truman and Ball, 1998*). Moreover, these conserved sets of NBs produce the neurons for

both the larval and adult CNS (**Prokop and Technau, 1991**). The duration of embryonic neurogenesis, however, differs in the two groups. Insects with direct development, like grasshoppers, produce all the neurons of the central brain and VNC during embryogenesis (**Shepherd and Bate, 1990**) and the hatchling possesses all the neurons of the adult (except for expansion of the Kenyon cell population, e.g., **Malaterre et al., 2002**). In insects with complete metamorphosis, by contrast, selection for the rapid formation of a larval stage required a premature arrest of neurogenesis, resulting in many fewer neurons for the larval brain (**Truman, 2005**). In *Drosophila*, for example, most brain and thoracic NBs produce only 10–15% of their respective progeny by hatching and the remainder are made during a second neurogenic period late in larval growth. Since essentially all their neurons are born during embryogenesis, the hatchlings of direct developing insects have brain circuits that include neurons generated during early, intermediate, and late phases of their neuroblasts' lineages. The larvae of metamorphic insects, by contrast, not only have fewer neurons to make their brain, but these neurons include only those with early-born fates.

A second key change involves the phenotypes of the neurons that are made during embryogenesis (**Figure 1**). The neurons of direct developing insects acquire their mature phenotype by the time of hatching and their anatomy and connections change very little through nymphal growth and adulthood (**Kutsch and Heckmann, 1995**). By contrast, in holometabolous insects, the form and function of many larval neurons are radically different from their adult form and function (e.g., **Truman and Reiss, 1976**; **Levine and Truman, 1985**; **Roy et al., 2007**). At metamorphosis, they lose their larval specializations and finally acquire their mature, adult phenotypes.

The analysis of development and metamorphosis of complex neuropils can provide important insights into the mechanisms underlying the formation of the larva and its subsequent metamorphosis (e.g., **Farnworth et al., 2020**). In our study, we have focused on the larval and adult circuitry of the mushroom bodies (MBs). In both stages, the MBs associate odors with either rewards or punishments and adjust the animal's future behavior accordingly (**Cognigni et al., 2018**; **Thum and Gerber, 2019**). The circuitry of the MB is known at the EM level for both the larva (**Eichler et al., 2017**) and adult (**Zheng et al., 2018**; **Li et al., 2020**). The core of the MB is a set of hundreds (larva) to thousands (adult) of small neurons called Kenyon cells (**Figure 2A and B**). Their dendrites project into calyx neuropil, which receives primarily olfactory input via antennal lobe projection neurons, and into accessory calyx neuropils that receive visual and thermal information (**Li et al., 2020**; **Eichler et al., 2017**). The bundled axons of the Kenyon cells extend down the peduncle and into the vertical and medial lobes. The mature, adult MB has three major classes of Kenyon cells: the γ neurons made during embryogenesis and early larval life, the α'β' cells generated late in larval life, and the αβneurons born through early and mid-metamorphosis (**Lee et al., 1999**). There is considerable complexity within each of these major Kenyon cell classes (**Eichler et al., 2017**; **Li et al., 2020**), but overall, their axons form three medial lobes (γ, β', β) and two vertical lobes (α, α') (**Figure 2B**). Unlike the adult, the MB of the larva contains only modified γ Kenyon cells whose axons form a vertical and a medial lobe (**Figure 2A**).

The MB receives flows of sensory information through projection neurons to the calyx, but our study focused on the sets of MB input neurons (MBINs) and output neurons (MBONs) that innervate the peduncle and lobes (**Figure 2A and B**). In both larvae and adults, these neurons divide the lobes into non-overlapping compartments (**Figure 2C**) that have a common microcircuit motif (**Figure 2D**; **Eichler et al., 2017**; **Zheng et al., 2018**). Each compartment is defined by the axonal tuft of an aminergic input cell that synapses onto Kenyon cell axons and onto a dedicated MBON(s). The Kenyon cell axons synapse onto each other and the MBONs but also feed back onto the MBINs (**Figure 2D**). The majority of the MBINs are dopaminergic neurons (DANs) but a few are octopaminergic neurons (OANs). Most DANs come from two clusters, the protocerebral anterior medial (PAM) cluster, which primarily encodes reward, and the protocerebral posterior lateral 1 (PPL1) cluster, which mainly encodes punishment (**Saumweber et al., 2018**; **Cognigni et al., 2018**; **Eichler et al., 2017**; **Eschbach et al., 2020**). Depending on their compartment, the MBONs are cholinergic, GABAergic, or glutamatergic. Interestingly, stimulation of the MBONs from PPL1-supplied compartments generally evokes approach behavior while stimulation of those from PAM supplied compartments evokes avoidance (**Owald et al., 2015**; **Cognigni et al., 2018**). Consequently, pairing punishment with a particular odor reduces the drive on MBONs that promote attraction. Behavior is therefore guided by a balance of avoidance versus attractive influences and the inhibition of neurons mediating one behavior then

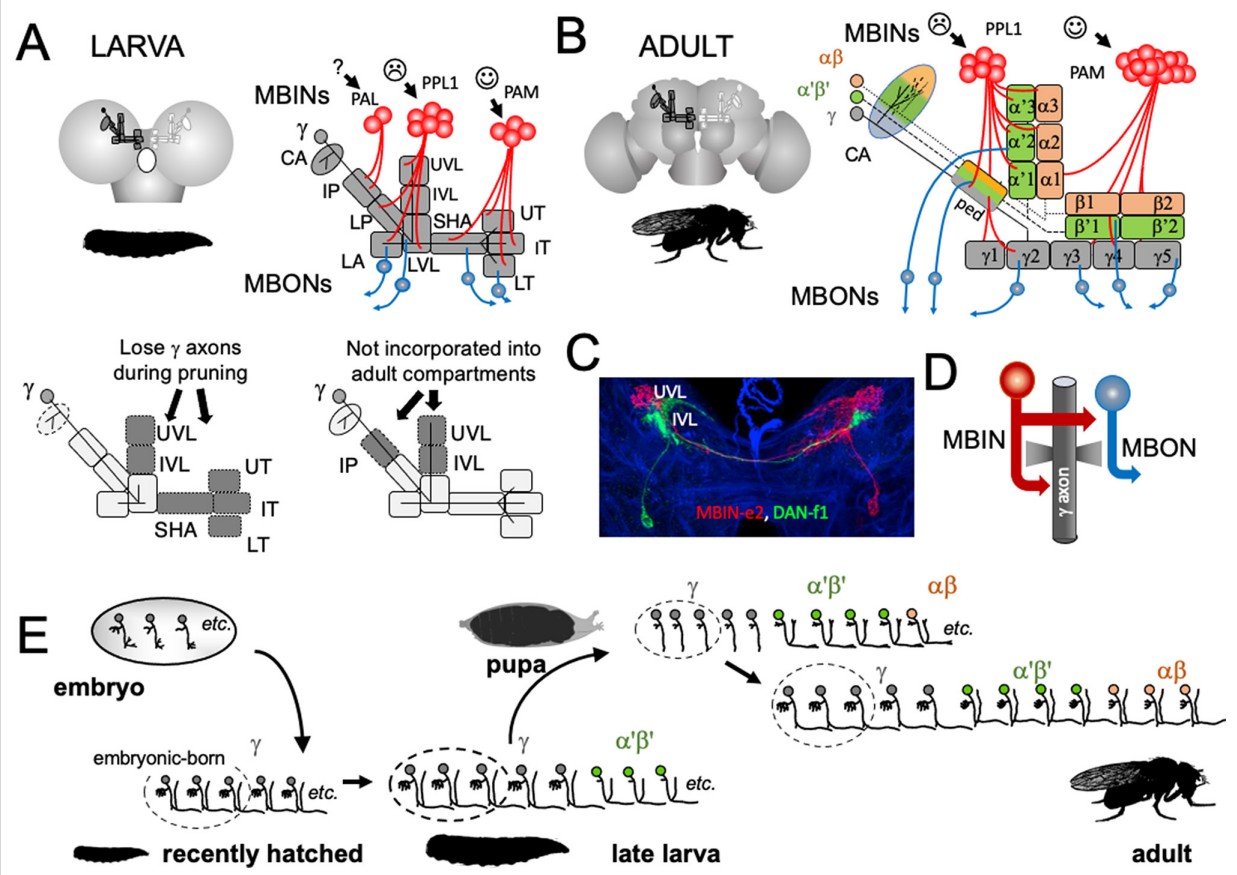

**Figure 2.** The organization and development of the larval and adult mushroom bodies (MBs). (**A**) The larval MB has a core of γ Kenyon neurons whose dendrites project to the calyx (CA) neuropil and whose axons extend through the peduncle and bifurcate into a vertical and medial lobe. Projections from three clusters of aminergic neurons, the PAL, PPL1, and PAM clusters divide the axon array into 10 computational compartments: IP and LP: intermediate and lower peduncle; LA: lateral appendix; UVL, IVL, LVL: upper, intermediate, and lower vertical lobe; SHA: shaft; UT, IT, LT: upper, intermediate, and lower toe. PPL1 input largely indicates punishment, PAM input indicates reward, and PAL is unknown. The diagrams below highlight in gray (left) the compartments that lose contact with γ neuron axons during pruning and (right) the compartments that are not incorporated into the adult MB. (**B**) The adult MB has 16 compartments. It contains regrown γ neurons (gray) that lack the larval-specific vertical branch along with late developing α'β' (green) and αβ (orange) Kenyon cells. These together form the medial (β',β) and vertical (α', α) lobe systems of the adult. Compartment designations are numbered and based on the Kenyon cell axons that they contain. (**C**) Projection of a multicolor flip-out (MCFO) image from a larval brain showing two MB input neurons that project bilaterally to the upper (UVL) and intermediate (IVL) compartments of the vertical lobes. Blue: neuroglian staining. (**D**) Schematic of the microcircuitry characteristic of larval and adult compartments. (**E**) Developmental timeline of the production of the three major classes of Kenyon cells that make up the mature MB.

The online version of this article includes the following source data for figure 2:

**Source data 1.** Examples of the adult anatomies of larval neurons MBIN-b1 and -b2 obtained by flip-switch-mediated immortalization of expression of line SS21716 late in larval life.

**Source data 2.** Examples of the adult anatomies of larval neurons DAN-c1 and DAN-d1 obtained by flip-switch-mediated immortalization of expression of lines MB586B and MB328B, respectively, late in larval life.

**Source data 3.** Examples of the adult anatomy of larval neuron OAN-e1 obtained by flip-switch-mediated immortalization of expression of lines SS21716 and SS01958 late in larval life.

**Source data 4.** Examples of the adult anatomies of larval neurons MBIN-l1 and DAN-f1 obtained by flip-switch-mediated immortalization of expression of stable spilt lines late in larval life.

**Source data 5.** Examples of the adult anatomies of larval neurons DAN-g1 and OAN-g1 obtained by flip-switch-mediated immortalization of expression of stable spilt lines late in larval life.

**Source data 6.** Table showing the success rate for maintaining expression of the various larval neurons through metamorphosis.

favors the opposite behavioral state (*Thum and Gerber, 2019*; *Cognigni et al., 2018*). The functions of some compartments, though, are complex because of extensive interconnections amongst MBONs and feedback from MBONs back to various MBINs (e.g., *Eschbach et al., 2020*; *Li et al., 2020*).

While serving similar functions of mediating associative learning, the larval and adult MB differ in a couple of ways. The larval MB has 10 compartments plus the calyx (*Saumweber et al., 2018*; *Eichler et al., 2017*), while the adult has 16 (*Aso et al., 2014*; *Figure 2A and B*). The larval structure lacks the α'β' and αβ neurons that form the adult system's vertical lobes. The larval γ neurons, though, have larval-specific vertical axon branches that form the core of a larval vertical lobe. We have focused on the MBINs and MBONs that establish the 10 larval compartments (*Eichler et al., 2017*; *Saumweber et al., 2018*). We have determined the metamorphic fates of about 80% of these neurons. Depending on the fate of their compartment, some larval neurons remain with the MB, others reprogram to join other adult circuits, and still others die. The persistence of MBON to MBIN connections would have been the simplest way that a larval memory trace could be carried through metamorphosis. However, we find that the diverse fates of the larval MB neurons plus the addition of new, adult-specific MBINs and MBONs to the compartments result in no MBIN-MBON pairings that survive from larva to adult.

## Results
### Metamorphic fates of the larval MBINs and MBONs

*Armstrong et al., 1998* used a set of enhancer-trap lines to follow subsets of extrinsic and intrinsic MB neurons through metamorphosis and showed that some of the medial lobe neurons functioned in both the larval and adult structures. Our use of a large collection of split-GAL4 lines that express in specific larval MBINs and MBONs (*Saumweber et al., 2018*) and a conditional flip-switch strategy (*Harris et al., 2015*) have allowed us to establish the metamorphic fates of most of the MBONs and MBINs. We focused on the larval unicompartmental neurons that possess the well-defined dendritic or axonal 'tufts' that define compartments (*Saumweber et al., 2018*), and we could determine the fates of 28 of the 32 classes of such cells. We have classified the persisting neurons as 'remodeling' if they continue as part of the MB circuitry after metamorphosis or as 'trans-differentiating' (see *Veverytsa and Allan, 2013*) if they retract from the MB system and function in other circuits of the adult brain. These two designations, however, likely are the two extremes along a continuum of change.

*Figure 3* and *Table 1* summarize the fates of the larval MBINs (*Saumweber et al., 2018*). We lacked suitable lines for the two octopaminergic neurons that innervate the calyx compartment (OAN-a1 and -a2). However, these cells have a very similar anatomy to the two adult OA-VUM2a neurons (*Busch et al., 2009*), leading us to conclude that they are the same neurons. We experimentally established the fates of 13 of the remaining 14 larval MBINs. These cells either trans-differentiate, remodel, or degenerate. Trans-differentiation was the fate of one MBIN innervating the LA compartment and all those innervating the vertical lobe compartments, UVL and IVL, and the intermediate peduncle (IP) compartment (*Figure 3*). The most extreme change was evident for MBINs-b1 and -b2; they withdraw from the larval intermediate peduncle and become sexually dimorphic neurons that innervate the adult optic lobes (*Figure 3B*, *Figure 3—figure supplement 1A and B*). We named their adult form PAL-OL because they are found in the protocerebral anterior lateral (PAL) cluster of adult aminergic neurons described by *Mao and Davis, 2009*. The three vertical lobe MBINs (OAN-e1, MBIN-e2, and DAN-f1) are members of the PPL1 group (*Saumweber et al., 2018*). We lacked a line to determine the fate of MBIN-e2, but OAN-e1 reorganizes to innervate the neuropil shell surrounding the MB lobes (as PPL1-SMP; *Figure 3—figure supplement 1C*) and DAN-f1 forms bilateral arbors in the adult superior medial protocerebrum as PPL1-bi-SMP (*Figure 3—figure supplement 1D*). The other trans-differentiating MBIN is MBIN-l1 that provides input from the larval lateral accessory lobe to the LA compartment of the MB; its adult form as LAL>bi-CRP redirects its lateral accessory lobe input to the crepine neuropil in both hemispheres (*Figure 3B*, *Figure 3—figure supplement 1E*).

Four other members of the PPL1 cluster (DAN-c1, -d1, and -g1, and MBIN-c2) innervate compartments at the base of the lobes and the peduncle (LVL, LP, and LA compartments). They undergo moderate remodeling and continue innervating compartments at the base of the mature MB peduncle and lobes (the PED, γ1, γ2, and α'1 compartments) (*Figure 3*). The octopaminergic MBIN, OAN-g1, exclusively targets the LVL compartment in the larva, but in its adult form, named OA-VPM3 (*Busch et al., 2009*), it extends extensive arbors into the fan-shaped body and the medial and lateral

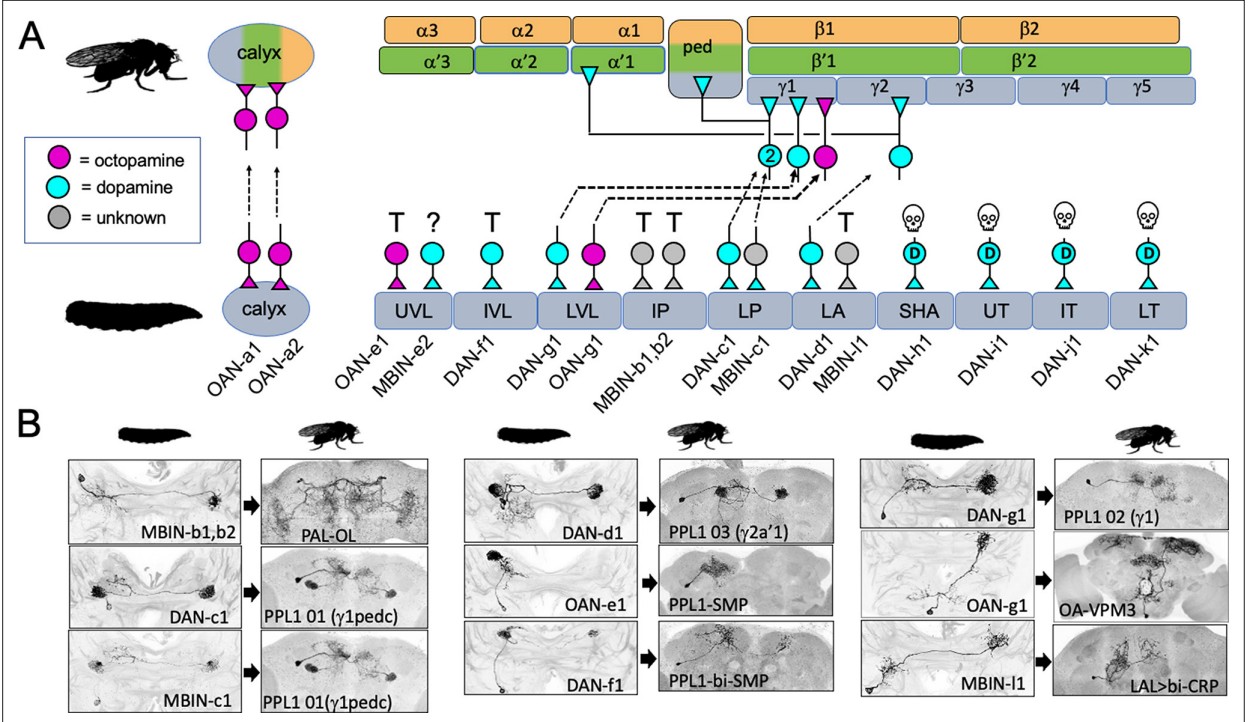

**Figure 3.** The metamorphic fates of the larval mushroom body input neurons (MBINs). (**A**) The fates of the larval MBINs that innervate the calyx and the 10 compartments of the larval MB. For larval MBINs that remain with the MB, the arrows show the relationship of their larval compartment to the one that they innervate in the mature, adult MB. The remaining MBINs die (skull), trans-differentiate (T) to supply non-MB circuits in the adult, or their fate is unknown (?). For the MBINs whose transmitter is unknown, they express tyrosine hydroxylase but their final secreted transmitter has not been determined. Compartment designations as in *Figure 2*. (**B**) Images comparing the larval and adult forms of the MBINs that persist through metamorphosis. The images of larval cells from *Saumweber et al., 2018* Nature Comm. 9: 1104. Adult names based on *Aso et al., 2014*, *Li et al., 2020*, or this study.

The online version of this article includes the following source data and figure supplement(s) for figure 3:

**Source data 1.** Examples of the adult anatomy of larval neuron MBON-a1 obtained by flip-switch-mediated immortalization of expression of lines SS01417 and SS00867 late in larval life.

**Source data 2.** Examples of the adult anatomies of larval neurons MBON-a2 and MBON-b1,-b2 obtained by flip-switch-mediated immortalization of expression of stable spilt lines late in larval life.

**Source data 3.** Examples of the adult anatomies of larval neurons MBON-d1, MBON-e2, and MBON-f2 obtained by flip-switch-mediated immortalization of expression of stable spilt lines late in larval life.

**Source data 4.** Examples of the adult anatomies of larval neurons MBON-g1 and -g2 obtained by flip-switch-mediated immortalization of expression of lines SS02130 and SS02121 late in larval life.

**Source data 5.** Examples of the adult anatomies of larval neurons MBON-h1 and -h2 obtained by flip-switch-mediated immortalization of expression of line SS01725 late in larval life.

**Source data 6.** Examples of the adult anatomies of larval neurons MBON-j1 and MBON-j2 obtained by flip-switch-mediated immortalization of expression of lines SS01973 and SS00860 late in larval life.

**Source data 7.** Examples of the adult anatomies of larval neurons MBON-i1 and MBON-k1 obtained by flip-switch-mediated immortalization of expression of lines SS01962 and SS04236 late in larval life.

**Figure supplement 1.** Confocal projections showing the terminal, adult identity of larval mushroom body input neurons (MBINs) that undergo trans-differentiation at metamorphosis.

**Figure supplement 2.** Confocal projections showing the terminal, adult structure of the larval neuron OAN-g1.

superior protocerebrum (*Figure 3—figure supplement 2*) with some arbor extending into adult γ1 compartment. This neuron shows profound metamorphic changes, but we still classify it as remodeling because it has some contact with the adult MB.

The four larval dopaminergic neurons of the PAM cluster innervate the four medial lobe compartments (SHA, UT, IT, and MT). Using our flip-switch method, we could not find an adult counterpart for

**Table 1.** Metamorphic fates of larval mushroom body extrinsic neurons.

| Larval name | Compartment | Lineage * | Adult identity | Ref for adult identity |
|---|---|---|---|---|
| **MBINs** | | | | |
| OAN-a1,a2 | CX | VUM[†] | OA-VUM2a | *Busch et al., 2009* |
| MBIN-b1,b2 | IP | DPLd | PAL-OL | *Mao and Davis, 2009*; this study |
| DAN-c1 | LP | CPd2/3 | PPL1 01 (γ1pedc) | *Aso et al., 2014*; *Li et al., 2020* |
| MBIN-c1 | LP | CPd2/3 | PPL1 01 (γ1pedc) | *Aso et al., 2014*; *Li et al., 2020* |
| DAN-d1 | LA | CPd2/3 | PPL1 03 (γ2α'1) | *Aso et al., 2014*; *Li et al., 2020* |
| MBIN-l1 | LA | BLV a3/4 | LAL>bi-CRP | This study |
| OAN-e1 | UVL | CPd2/3 | PPL1-SMP | *Mao and Davis, 2009*; this study |
| MBIN-e2 | UVL | CPd2/3 | Unknown | |
| DAN-f1 | IVL | CPd2/3 | PPL1-bi-SMP | *Mao and Davis, 2009*; this study |
| DAN-g1 | LVL | CPd2/3 | PPL1 02 (γ1) | *Aso et al., 2014*; *Li et al., 2020* |
| OAN-g1 | LVL | Unknown | OA-VPM3 | *Busch et al., 2009* |
| DAN-h1 | SHA | DAL CM-1/2 | Dead | This study |
| DAN-i1 | UT | DAL CM-1/2 | Dead | This study |
| DAN-j1 | IT | DAL CM-1/2 | Dead | This study |
| DAN-k1 | LT | DAL CM-1/2 | Dead | This study |
| **MBONs:** | | | | |
| MBON-a1 | CX | CPv2/3 | MBON 29 (γ4γ5) | This study |
| MBON-a2 | CX | CPv2/3 | MBON 22 (calyx) | *Aso et al., 2014*; *Li et al., 2020* |
| MBON-b1,-b2 | IP | BLVa3/4 | LH-LN | *Dolan et al., 2019* |
| MBON-b3 | IP | CPv2/3 | Unknown | |
| MBON-c1 | LP | BLDc | MBE-CA | This study |
| MBON-d1 | LA | DAL CM-1/2 | MBON 11 (γ1pedc>α/β) | *Aso et al., 2014*; *Li et al., 2020* |
| MBON-d2 | LA | BAmd2 | SMP>IB | This study |
| MBON-e1 | UVL | CPd2/3 | Unknown | |
| MBON-e2 | UVL, IVL, LVL | DAM-d1 | MBON 03 (β'2mp) | *Aso et al., 2014*; *Li et al., 2020* |
| MBON-f2 | IVL | DAL cl2 | MBON 30 (γ1,γ2,γ3) | *Li et al., 2020* |
| MBON-f1 | IVL | CPd | Unknown | |
| MBON-g1,g2 | LVL | DAL-V2/3 | LAL.s-NO$_2$i.b | *Wolff and Rubin, 2018* |
| MBON-h1 | SHA | DAL-V2/3 | MBON 09 (γ3β'1) | *Aso et al., 2014*; *Li et al., 2020* |
| MBON-h2 | SHA | DAL-V2/3 | MBON 08 (g3) | *Aso et al., 2014*; *Li et al., 2020* |
| MBON-i1 | UT | DAM-d1 | MBON 04 (β'2-bilat) | *Aso et al., 2014*; *Li et al., 2020* |
| MBON-j1 | IT | DAM-d1 | MBON 02 (β2β'2a) | *Aso et al., 2014*; *Li et al., 2020* |
| MBON-j2 | IT | DAL CM-1/2 | MBON 05 (γ4>γ1,γ2) | *Aso et al., 2014*; *Li et al., 2020* |
| MBON-k1 | LT | DAM-d1 | MBON 01 (γ5β'2a) | *Aso et al., 2014*; *Li et al., 2020* |
| APL | UT,LT,LA,VL,CX | BLV a3/4 | APL | *Aso et al., 2014*; *Li et al., 2020* |

CX, calyx; IP: intermediate peduncle; LP: lower peduncle; UVL: upper vertical lobe; IB: inferior bridge; IVL: intermediate vertical lobe; LVL: lower vertical love; LA: lateral appendix; OT: optic tubercle; SHA: shaft; SMP: superior medial protocerebrum; UT: upper toe; IT: intermediate toe; LT: lower toe.

*Lineage designations from *Saumweber et al., 2018*.

[†]Lineage assumed to be from the ventral unpaired neuroblast because of position and nature of neurons.

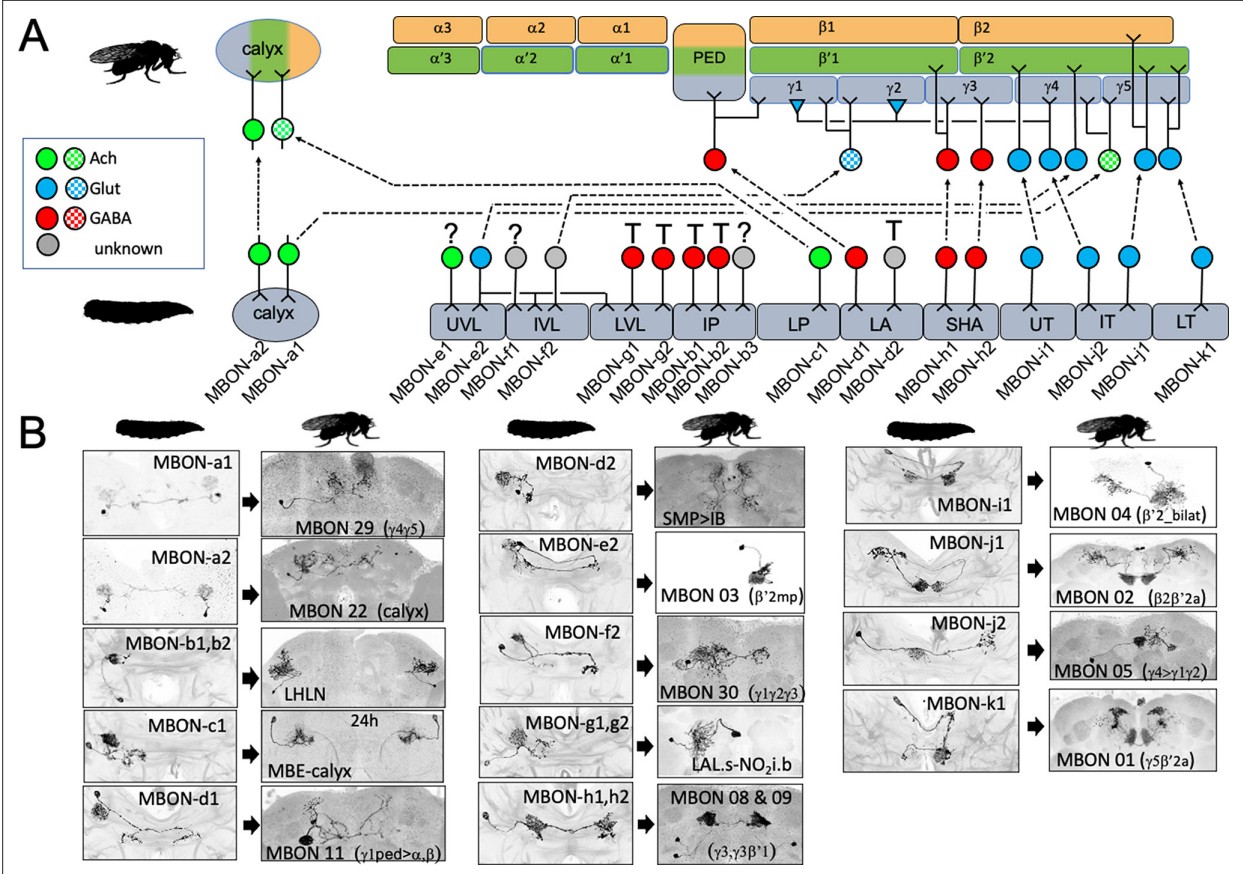

**Figure 4.** The metamorphic fates of the larval mushroom body output neurons (MBONs). (**A**) The fates of the larval MBONs that innervate the calyx and the 10 compartments of the larval MB. For MBONs that remain within the MB after metamorphosis, the arrows show the relationship of their larval compartment to the one that they innervate in the adult MB. The remaining MBONs trans-differentiate (T) to supply non-MB circuits in the adult, or their fate is unknown (?). Compartment designations as in *Figure 2*. Transmitters: green: acetylcholine; blue: glutamate; red: GABA; gray: unknown; checkered versions are presumed transmitters based on *Li et al., 2020*. (**B**) Images comparing the larval and adult forms of the MBONs that persist through metamorphosis. The images of larval cells from *Saumweber et al., 2018 Nature Comm*. 9: 1104. Adult names based on *Aso et al., 2014*, *Li et al., 2020*, or this study.

The online version of this article includes the following source data and figure supplement(s) for figure 4:

**Source data 1.** Examples of the adult anatomies of larval neuron APL obtained by flip-switch-mediated immortalization of expression of line SS01671 late in larval life.

**Figure supplement 1.** Confocal projections showing the terminal, adult identity of larval mushroom body output neurons (MBONs) that undergo trans-differentiation at metamorphosis.

any of them despite using nine different split-GAL4 lines. We therefore assumed that they died during metamorphosis (*Figure 3A*) and this is confirmed in the next section.

We established the fates of 14 of the 17 types of larval MBONs (*Figure 4*, *Table 1*). None died. They either remodeled or trans-differentiated, although as described for OAN-g1 above, such a distinction is not clear-cut for some neurons. We categorized the two calyx neurons, MBON-a1 and -a2, as remodeling because they also innervate the mature MB. This is clearly the case for MBON-a2 since its larval and adult morphologies (as MBON 22) are extremely similar. MBON-a1, by contrast, completely withdraws from the calyx and directs its adult growth into the adult γ4 and γ5 compartments as MBON 29. Three larval MBONs from the lobe system show similar dramatic morphological changes but also remain within the MB circuit. MBON-c1 retracts from the larval lower peduncle compartment and grows into the adult calyx (as MBE-calyx). MBON-e2 and MBON-f2 retract from the vertical lobes and regrow into medial lobe compartments, becoming the adult MBON 03 and MBON 30, respectively. The MBONs innervating the larval medial lobe compartments (SHA, UT, IT, and LT)

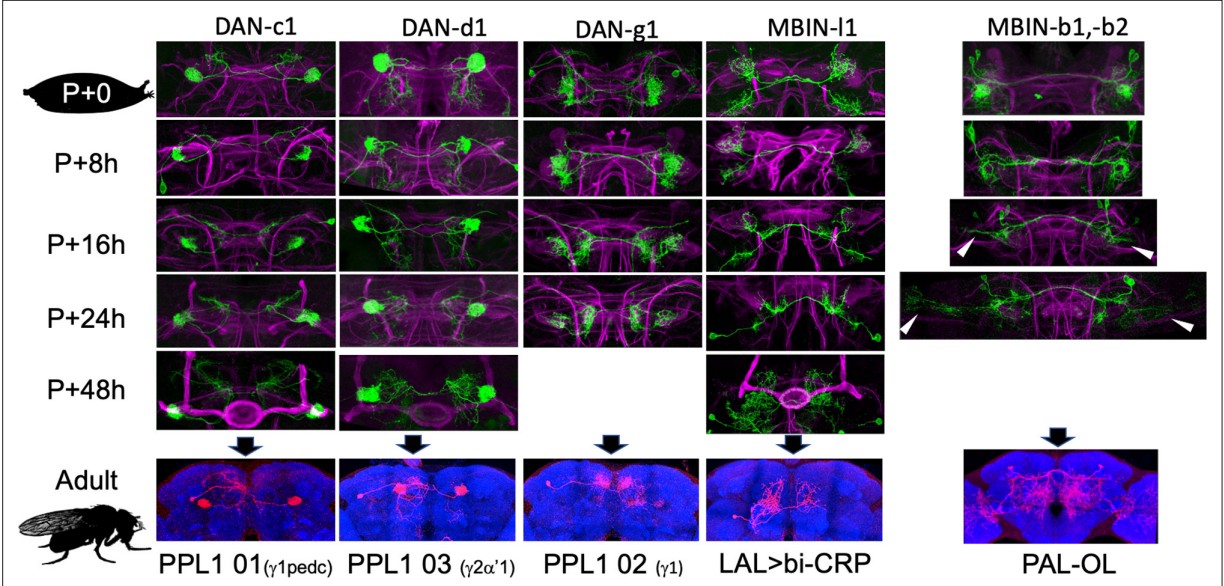

**Figure 5.** The metamorphic transformation of selected mushroom body input neurons (MBINs) during early metamorphosis and their mature phenotype in the adult. Confocal images track the GFP expression through the first 48 hr after pupariation (P); the adult images show flip-switch induced expression of red fluorescent protein. Arrowheads: growth cones; P+#: # hours after pupariation. Background staining for the developmental series is for Fasciclin II (magenta); for the adult, it is Bruchpilot (nc82) (blue). Lines used for developmental timelines: DAN-c1: JRC-SS03066, DAN-d1: JRC-MB328B, DAN-g1: JRC-SS01716, MBIN-l1: JRC-SS04484, MBIN-b1,-b2: JRC-SS21716.

The online version of this article includes the following figure supplement(s) for figure 5:

**Figure supplement 1.** Confocal images following the degeneration of GFP-labeled larval PAM neuron early in metamorphosis.

**Figure supplement 2.** Confocal images showing the early metamorphic changes in line JRC-SS01702 driving GFP expression in two mushroom body input neurons (MBINs): OAN-e1 and DAN-c1.

are more conservative in their changes; they persist as medial lobe MBONs and supply topologically similar compartments in the adult (*Figure 3*; MBON-h1 to -k1).

As with the MBINs, the MBONs that trans-differentiated were typically those from larval vertical lobe compartments and the intermediate peduncle (*Figure 4A and B*). For example, MBON-b1 and -b2 retract from the larval IP compartment and invade the lateral horn where they become local inter-neurons (*Figure 4*, *Figure 4—figure supplement 1B*; LHLN neurons; *Dolan et al., 2019*). The most striking changes occur in MBON-g1 and -g2 (*Figure 4B*, *Figure 4—figure supplement 1C and D*), which transform into central complex neurons – the LAL.s-NO2i.b neurons that innervate the nodulus (*Wolff and Rubin, 2018*).

### The time course of MBIN and MBON metamorphic changes

The fates of most of the larval MBINs and MBONs, as determined by the flip-switch method, were confirmed by following the expression of the parental lines through early metamorphosis. Although most enhancer-based lines change their expression patterns during metamorphosis, we found that GFP expression typically persists through enough of the remodeling period to confirm a neuron's adult identity.

As expected from our failure to find the adult versions of the medial lobe MBINs (DAN-h1 to -k1) using the flip-switch method, we found that they degenerate early in metamorphosis (*Figure 5—figure supplement 1*). Their dendritic arbors collapsed by 8 hr after puparium formation (APF), the cell bodies were disrupted by 16 hr APF, and the neurons were reduced to scattered GFP-labeled debris by 24 hr APF.

The MBINs and MBONs that either remodel or trans-differentiate (*Figures 5 and 6*) showed a time course of pruning and outgrowth like that reported for the γ Kenyon cells (*Watts et al., 2003*). Pruning of γ neuron axons is evident by 4 hr APF and is completed by 16–18 hr APF. Adult outgrowth commences by 24 hr and is finished by 48 hr APF (*Yaniv et al., 2012*; *Mayseless et al., 2018*). The larval MBINs (*Figure 5*, *Figure 5—figure supplement 2*) and MBONs (*Figure 6*) also showed arbor

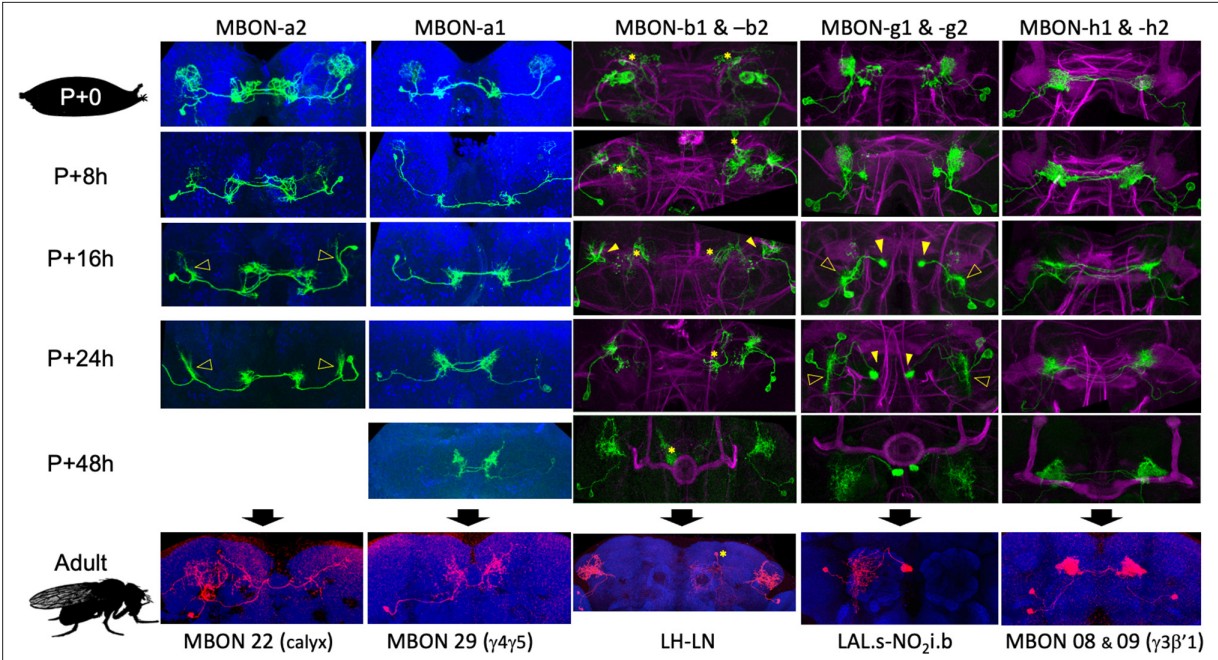

**Figure 6.** The metamorphic transformation of selected mushroom body output neurons (MBONs) during early metamorphosis and their mature phenotype in the adult. Confocal images track the GFP expression through the first 48 hr after pupariation (P); the adult images show flip-switch-induced expression of red fluorescent protein. *Expression due to nontarget neurons in some driver lines; filled arrowheads: axonal growth cones; open arrowheads: dendritic growth cones; P+#: # hours after pupariation. Magenta: Fasciclin II; blue: Bruchpilot (nc82). Lines used for developmental timelines: MBON-a1: JRC-SS00867, MBON-a2: JRC-SS02006, MBON-g: JRC-SS02130, MBON-h: JRC-SS01725, MBON-j1: JRC-SS01973.

The online version of this article includes the following figure supplement(s) for figure 6:

**Figure supplement 1.** The early metamorphic transformation in selected larval mushroom body output neurons (MBONs) that assume similar phenotypes in the adult system.

**Figure supplement 2.** Pruning and outgrowth of MBON-j2 as it transforms into its adult form named MBON 05.

**Figure supplement 3.** The early metamorphic transformation of larval mushroom body output neurons (MBONs) that show major redirections to adult mushroom body compartments.

loss by 8 hr APF and were maximally pruned by 16 hr APF. Growth cones were evident between 16 and 24 hr APF. MBIN-b1 and -b2 showed the most exuberant outgrowth, having formed growth cones that were halfway to the optic lobes by 16 hr APF and had reached them by 24 hr (*Figure 5*). Most neurons achieved their adult form by 48 hr APF. The GABAergic anterior paired lateral (APL) neuron and MBON-j2 neurons were exceptional in that they continue arbor extension beyond 48 hr (*Figure 6—figure supplements 1 and 2*). APL eventually covers all of the adult compartments by 72 hr APF (*Mayseless et al., 2018*) and is involved in feedback suppression in both larva and adult (*Liu and Davis, 2009*; *Masuda-Nakagawa et al., 2014*; *Saumweber et al., 2018*). For MBON-j2, its γ4 tuft forms at the same time as those of other MBONs but the formation of its γ2 and especially its γ1 arbors is delayed. (*Figure 6—figure supplement 2*). In its adult function as MBON 05 (AKA MBON-γ4>γ1,γ2), it is suggested to provide feed-forward inhibition between compartments (*Aso et al., 2014*). Its extended period of outgrowth may allow time for the compartment microcircuits to become established before it interconnects them.

MBON-a1 and -a2 provide an interesting contrast of divergent remodeling of two similar larval cells (*Figure 6*). Both cells show retraction of dendritic and axonal arbors by 8 hr APF. MBON-a1 removes its larval dendritic arbor by 16 hr, and its distal axonal region then extends new growth cones to innervate the adult γ4 and γ5 compartments and surrounding neuropils. In contrast, the dendritic arbor of MBON-a2 only partially regresses. It organizes into a dendritic growth cone by 16 hr APF, which then reinvades the calyx. Its adult form, MBON-20 (MBON-calyx; *Aso et al., 2014*; *Li et al., 2020*), is very similar to its larval form.

The most stable extrinsic neuron is DAN-d1 (*Figure 5*). Its dendritic arbor remains intact through metamorphosis. Its axonal tuft shows moderate thinning from 8 to 18 hr APF and then extends fine processes into the forming γ2 and α'1 compartments between 24 and 48 hr APF. A more extreme remodeling is evident in the larval MBON-d1, which becomes the adult cell MBON 11 (*Figure 6—figure supplement 1*). Its larval dendritic tuft retracts and reorganizes into a growth cone by about 18 hr APF. The latter extends into the γ1 and peduncle compartments by 24 hr and achieves its adult dendritic configuration by 48 hr APF. Its axonal arbor changes from a compact projection in the larva to a sparse arbor innervating the adult α and β lobes (*Aso et al., 2014*; *Li et al., 2020*).

Amongst the cells undergoing trans-differentiation, the most extreme changes are seen in MBON-g1 and g2, which shift from the larval MB to the adult central complex (*Figure 6*). Their larval axonal arbors are retracting by 8 hr APF and organize into axonal growth cones by 18 hr APF. The growth cones then navigate medially to innervate the intermediate zone of the nodulus. The dendritic tufts of MBON-g1 and -g2 thin by 8 hr APF and then fragment as new dendritic growth cones sprout from the base of the old arbor (18 hr APF). The dendrites innervate the ipsilateral lateral accessory lobe of the adult.

MBON-c1 was refractory to the flip-switch approach, but its early metamorphic changes gave us insight into its mature function. As seen in *Figure 6—figure supplement 3*, by 24 hr APF, its larval dendritic arbor is essentially gone, and new growth cones are invading the calyx neuropil. This split-GAL4 line lost its GFP expression after 24 hr, but the extensive invasion of the calyx neuropil by this time indicates that the calyx is its adult target. Its anatomy at 24 hr APF, though, is too immature

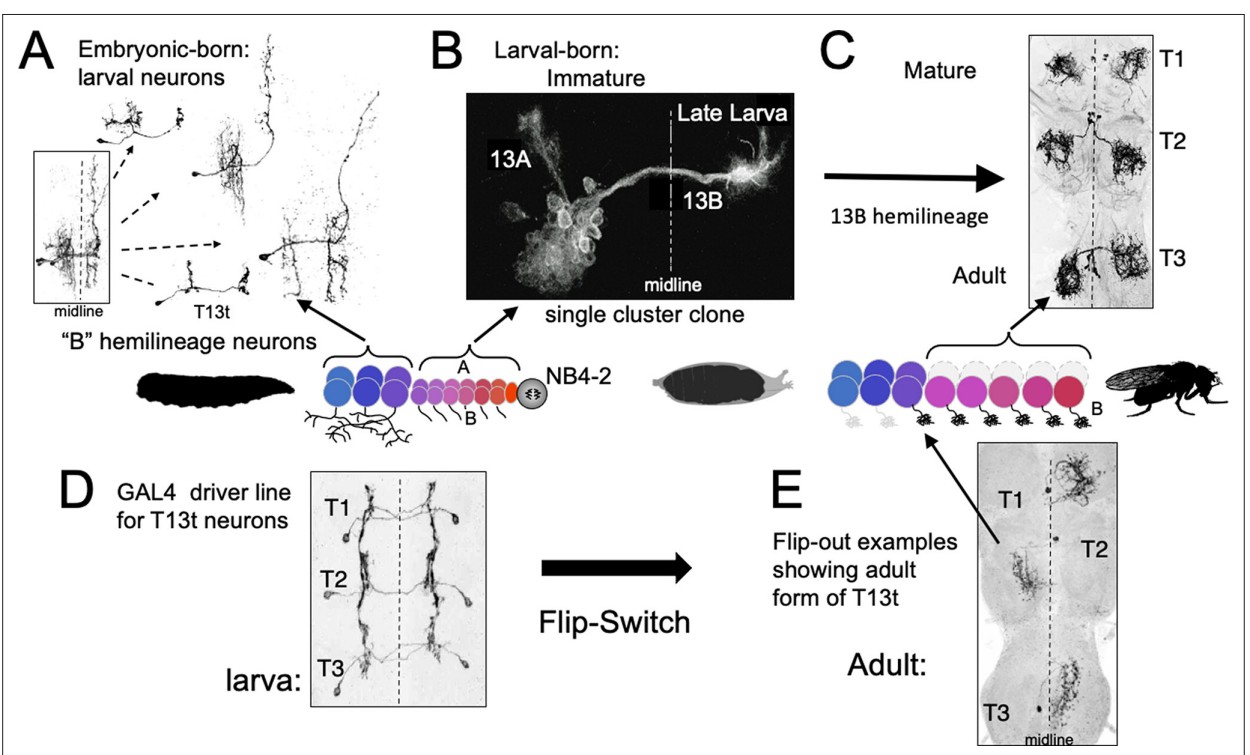

**Figure 7.** The phenotypes of neurons made by thoracic neuroblast (NB) 4–2 during its embryonic and postembryonic phases of neurogenesis. Neurons are generated pairwise during both phases to produce the 13A and 13B hemilineages. (**A**) Four examples of embryonic born, 13B interneurons that function in the larva. All are commissural interneurons having ipsilateral dendrites and contralateral output sites. Boxed image shows the neurons overlapping in segment T1. (**B**) Example of a postembryonic clone of NB4-2 showing the neurons of the two hemilineages at the end of larval life. (**C**) Confocal projection of the adult form of larval-born, 13B interneurons expressed in the SS04739 driver line. (**D, E**) The larval and adult phenotypes of one of the embryonic born 13B interneurons, T13t. (**D**) shows the three pairs of T13t neurons as revealed by the SS02006 driver line. (**E**) shows Flip-switch clones of the same cells showing their adult phenotypes.

The online version of this article includes the following figure supplement(s) for figure 7:

**Figure supplement 1.** Confocal images of dorsal (top) and transverse (bottom) views of the early metamorphosis of the T3 pair of T13t cells: the dendritic arbor is gone by 8 hr after pupariation (*P+8h*), contralateral growth cones are evident by *P+24h*, and the arbor is near its maximal extent by *P+48h*.

to allow it to be matched to any described adult cell, so we have called its adult form MBE-CA (mushroom body extrinsic neuron to calyx).

## The larval form is a derived state for the neurons that show trans-differentiation

The γ Kenyon cells first assume a larval form, with vertical and medial axon branches, and then remodel into their adult form. The latter form, with its single medially directed axon, is the same as seen in the γ neurons of nymphs and adults of the more basal, direct-developing insects, such as the cricket (*Malaterre et al., 2002*). Thus, the adult version of *Drosophila* γ neurons is like that of a direct developing ancestor, while its larval form is derived to accommodate the lack of αβ class neurons in the larva. Is this a general rule for cells that have different larval and adult phenotypes? Do the adult phenotypes approximate the ancestral phenotype while the larval phenotypes are derived? It is difficult to directly deal with these designations for the trans-differentiating MBONs and MBINs because we do not know the corresponding neurons in direct developing species. We can, however, make this comparison for some VNC neurons such as the thoracic midline spiking interneurons (*Figure 7*). These neurons are found in a large cluster in each thoracic hemineuropil, and each cluster is the progeny of a single, identified neuroblast (*Shepherd and Laurent, 1992*). Each cluster includes two sets of GABAergic neurons that are based on the asymmetrical division of the ganglion mother cells. The 'A' group (hemilineage) remains ipsilateral, while the 'B' hemilineage projects to the contralateral leg neuropil where, as the midline spiking interneurons, they shape the response of leg motoneurons to input from leg mechanoreceptors (*Siegler and Burrows, 1984*). They were first described in grasshoppers (*Siegler and Burrows, 1984*) but they are found in both direct developing and metamorphic insects (*Witten and Truman, 1998*), indicating their involvement in leg function through insect evolution.

In *Drosophila*, the corresponding interneurons are from the 13B hemilineage, which is produced by neuroblast NB4-2 (*Figure 7*; *Harris et al., 2015*; *Lacin and Truman, 2016*). The 'B' class of embryonic-born neurons from this lineage serve as commissural interneurons in the larva (*Figure 7A*), receiving ipsilateral input and having contralateral output. Their bundled, commissural axons provide a pathway that their postembryonic-born counterparts, the 13B interneurons, follow to the contralateral neuropil where they stop and wait for metamorphosis (*Figure 7B*). They then mature into local leg interneurons (*Figure 7C*) that have a form very different from their larval counterparts. This difference between the two sets of 13B neurons is only temporary, however. *Figure 7D and E* (*Figure 7—figure supplement 1*) follows the metamorphic fate of one of the embryonic-born, larval interneurons from this hemilineage. At metamorphosis, this neuron, T13t, loses its ipsilateral arbor and reorganizes its contralateral arbor to become a local leg interneuron like its postembryonic-born counterparts.

Therefore, despite the larva lacking legs, *Drosophila* starts making leg interneurons during embryogenesis, but these embryonic-born cells initially take on a phenotype adapted to the needs of the legless larva. Their ancestral functions as leg interneurons are only manifest at metamorphosis. Like the γ Kenyon cells, these neurons support the idea that for neurons that function in both larva and adult, their larval phenotype is a derived state adapted to the larval stage, while their adult phenotype is more in line with the ancestral function of the cell.

## Origins of adult-specific MBINs and MBONs

Remodeled larval MBINs and MBONs account for only 15 of the 41 different types of adult MBINs and MBONs (*Aso et al., 2014*). The remaining 26 types could either be born during the postembryonic neurogenic period or they could come from neurons that function outside of the MB circuit in the larva and then switch to the MB at metamorphosis. We could determine the origins of 22 of these 26 types (*Figure 8A and B*, *Table 2*) and all are born postembryonically.

The origins of the remaining adult MBINs have already been determined (*Figure 8A*). About 150 PAM neurons divided into at least 15 different types are found in the adult MB (*Aso et al., 2014*; see also *Lee et al., 2020*; *Li et al., 2020*). These neurons are born during the postembryonic phase of neurogenesis and come from the CREa1A and CREa2A neuroblasts (*Lee et al., 2020*; *Table 2*). For the PPL1 cluster MBINs, the number is about the same in the larva and the adult, but the cell population partially changes. We find that three of the larval PPL1 MBINs retract from the MB and join other circuits in the adult (*Figure 3*), and *Ren et al., 2016* found that three adult-specific neurons are added from the postembryonic DL2 lineage.

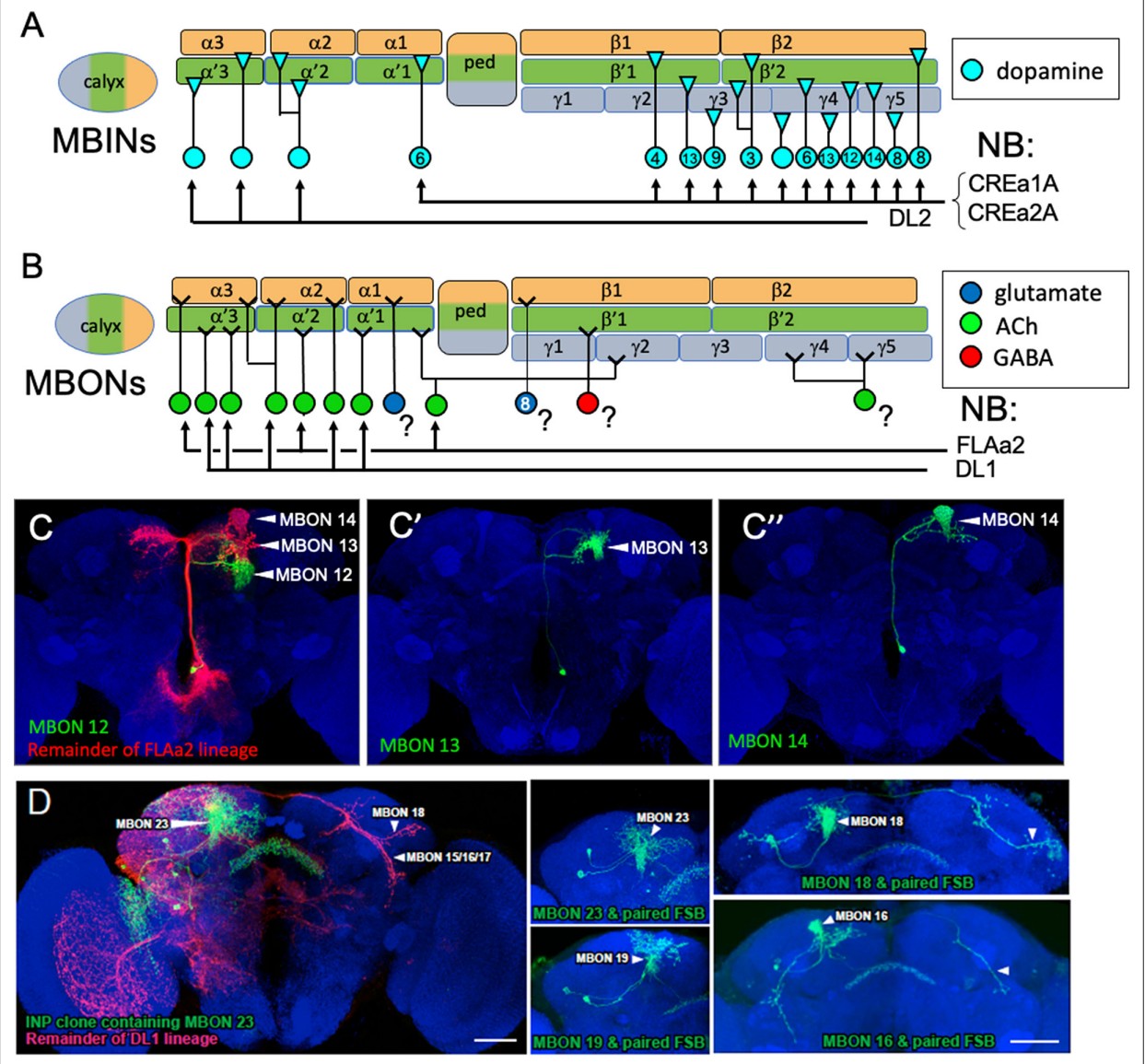

**Figure 8.** The postembryonic-born mushroom body input neurons (MBINs) and mushroom body output neurons (MBONs) of the adult mushroom bodies. (**A**) Summary of the origins of the adult MBINs that arise during the postembryonic period. Numbers give the number of neurons in each of the PAM groups (from *Aso et al., 2014*). (**B**) Summary of the origins of the adult MBONs that arise during the postembryonic period. NB: neuroblast; ?: adult MBONs whose origins are unknown. (**C**) Results of twin-spot MARCM approach showing the sequential postembryonic birth of FLAa2 lineage neurons that innervate α' and α compartments. (**C**), (**C'**), and (**C"**) images are produced by successively later heatshocks in the larva; green cells are produced after the heatshock while the red cells (shown only in **C**) are the remainder of the FLAa2 lineage. (**D**) Twin spot MARCM results from the type II DL1 lineage. The leftmost panel shows the progeny of an intermediate neural progenitor (INP) in green and the remainder of the lineage in red. The remaining panels show GMC clones with an MBON neuron and its paired sister fan-shaped body (FSB) neuron, both in green. The arbors identifying the individual MBONs are marked. Background staining (blue) is for Bruchpilot (nc82).

We determined the origins of nine of the thirteen remaining types of adult MBONs (*Table 2*, *Figure 7B*) using the twin-spot MARCM technique (*Yu et al., 2009*). As seen in *Figure 8C*, MBONs 12 (γ2α'1), 13 (α'2), and 14 (α3) are born in succession during the early postembryonic phase of the FLAa2 lineage. The remaining vertical lobe MBONs arise in the DL1 lineage (*Figure 8D*, *Table 2*). The DL1 neuroblast has a type II pattern of division (*Boone and Doe, 2008*; *Wang et al., 2014*). Each division produces an intermediate precursor cell, which, in turn, produces a small series of ganglion mother cells. Each of the latter divides to make two daughter neurons. As shown in *Figure 8D* for at least six of these GMCs, one daughter becomes a vertical lobe MBON while the other becomes a central complex neuron that innervates the fan-shaped body.

**Table 2.** Developmental origins of adult mushroom body input neurons (MBINs) and mushroom body output neurons (MBONs) that do not come from remodeled larval, extrinsic mushroom body neurons.

| Neuron | # | Origin | Lineage | Reference |
|---|---|---|---|---|
| **MBINs** | | | | |
| PPL1_04 (α'3) | 1 | Postembryonic | DL2 | *Ren et al., 2016* |
| PPL1-05 (α'2α2) | 1 | Postembryonic | DL2 | *Ren et al., 2016* |
| PPL1-06 (α3) | 1 | Postembryonic | DL2 | *Ren et al., 2016* |
| PAM 01 (γ5) | 19 | Postembryonic | CREa1A, CREa2A | *Lee et al., 2020* |
| PAM 02 (β'2a) | 8 | Postembryonic | CREa1A, CREa2A | *Lee et al., 2020* |
| PAM 03 (β2β'2a) | 4 | Postembryonic | CREa1A, CREa2A | *Lee et al., 2020* |
| PAM 04 (β2) | 16 | Postembryonic | CREa1A, CREa2A | *Lee et al., 2020* |
| PAM 05 (β'2p) | 10 | Postembryonic | CREa1A, CREa2A | *Lee et al., 2020* |
| PAM 06 (β'2m) | 15 | Postembryonic | CREa1A, CREa2A | *Lee et al., 2020* |
| PAM 07 (γ4<γ1γ2) | 5 | Postembryonic | CREa1A, CREa2A | *Lee et al., 2020* |
| PAM 08 (γ4) | 26 | Postembryonic | CREa1A, CREa2A | *Lee et al., 2020* |
| PAM 09 (β1ped) | 6 | Postembryonic | CREa1A, CREa2A | *Lee et al., 2020* |
| PAM 10 (β1) | 6 | Postembryonic | CREa1A, CREa2A | *Lee et al., 2020* |
| PAM 11 (α1) | 7 | Postembryonic | CREa1A, CREa2A | *Lee et al., 2020* |
| PAM 12 (γ3) | 11 | Postembryonic | CREa1A, CREa2A | *Lee et al., 2020* |
| PAM 13 (β'1ap) | 7 | Postembryonic | CREa1A, CREa2A | *Lee et al., 2020* |
| PAM 14 (β'1) | 8 | Postembryonic | CREa1A, CREa2A | *Lee et al., 2020* |
| PAM 15 (γ5β'2a) | 3 | Postembryonic | CREa1A, CREa2A | *Lee et al., 2020* |
| PAM γ4/5 | ? | Postembryonic | CREa1A, CREa2A | *Lee et al., 2020* |
| **MBONs** | | | | |
| MBON 06 (β1>α] | 1 | Unknown | | |
| MBON 07 (α1) | 2 | Unknown | | |
| MBON 10 (β'1) | 8 | Unknown | | |
| MBON 12 (γ2α'1) | 2 | Postembryonic | FLAa2 | This study |
| MBON 13 (α'2) | 2 | Postembryonic | FLAa2 | This study |
| MBON 14 (α3) | 2 | Postembryonic | FLAa2 | This study |
| MBON 15 (α'1) | 2 | Postembryonic | DL1 | This study |
| MBON 16 (α'3ap) | 1 | Postembryonic | DL1 | This study |
| MBON 17 (α'3m) | 2 | Postembryonic | DL1 | This study |
| MBON 18 (α2sc) | 1 | postembryonic | DL1 | This study |
| MBON 19 (α2p3p) | 2 | Postembryonic | DL1 | This study |
| MBON 21 (γ4,γ5) | 1 | Unknown | | |
| MBON 23 (α2sp) | 1 | Postembryonic | DL1 | This study |
| MB-DPM | 1 | Postembryonic | Unknown | *Mayseless et al., 2018* |

Adult names according to *Aso et al., 2014* and *Li et al., 2020*, except PAM γ4/5, which is based on *Lee et al., 2020*.

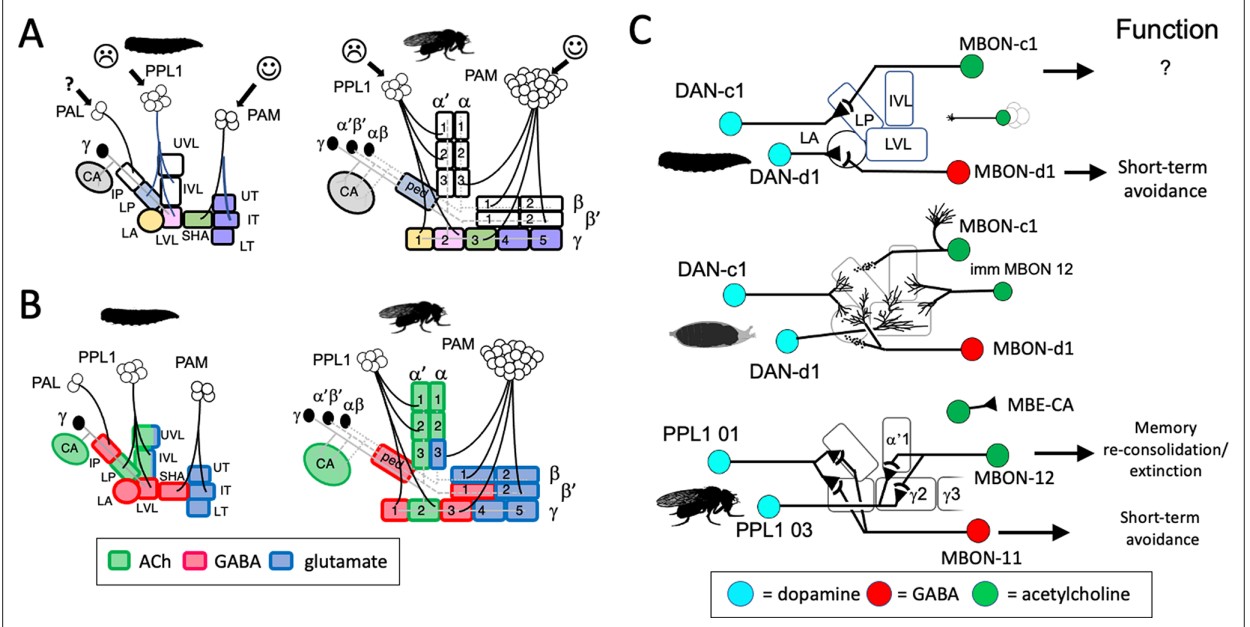

**Figure 9.** Stability and changes in mushroom body (MB) compartments during metamorphosis. (**A**) Developmental fates of the larval compartments through metamorphosis. Larval compartments that are lost at metamorphosis are uncolored. Those incorporated into the mature structure share the same color in the two stages. (**B**) Summary of transmitter output from the MB compartments in the larval and adult stages. The larval LP (=adult ped) and LVL (=adult γ 2) compartments switch transmitters through metamorphosis. Compartment designations as in *Figure 2*. (**C**) Summary of the roles of mushroom body output neuron (MBON) compartment shifting, MBON trans-differentiation, and MBON recruitment in producing the output configuration of the mature MB. Persisting neurons in the lobe compartments are DAN-c1/PPL1 01, DAN-d1/PPL1 03, and MBON-d1/MBON 11; the new adult-specific neuron is MBON 12; and MBON-c1/MBE-CA shifts to the calyx.

## Discussion

Some neurons like the GABAergic APL have a unique and characteristic morphology that allows them to be readily identified in both larval and adult stages and the same name has been used for both stages. The vast majority of MBINs and MBONs, however, have been given different names in the larva versus the adult (*Saumweber et al., 2018*; *Aso et al., 2014*), and this paper determines their correspondence for the first time. Where useful in the 'Discussion,' we will combine their larval and adult names. For example, since larval MBON-d1 becomes adult MBON 11, it will be referred to as MBON-d1/MBON 11.

### Metamorphosis of mushroom body compartments

The evolution of a metamorphic life history in insects required two changes: first, the modification of embryogenesis to produce a larval body; and second, metamorphosis itself –the transformation of that larval body into the insect's mature form. Our focus has been on the second process of transforming the larval MB into its adult form, but our results also provide insight into the first issue, i.e., how embryonic development may have been altered to form a specialized larval MB.

We focused on the metamorphic fates of the MBINs and MBONs that make the microcircuits that divide the MB lobes into their computational compartments. As summarized in *Figure 9A*, three compartments are specialized for the larval stage and not carried into the adult. For the two distal vertical lobe compartments, UVL and IVL, their γ axon core degenerates and two of their MBONs (MBON-e2 and MBON-f2) switch to adult medial lobe compartments while their remaining MBONs and MBINs shift to adult circuits outside of the MB (*Figures 3 and 4*). The larval vertical lobe is then replaced with new, adult vertical lobes formed from α' and α axon cores and postembryonic-born MBINs (*Figure 7A*) and MBONs (*Figure 7B*). There is, however, no cellular continuity between the larval vertical lobe and mature vertical lobes of the adult MB. We think of the larval structure as a vertical lobe 'facsimile'; a larval innovation to deal with the lack of the Kenyon cells and extrinsic neurons that typically make the vertical lobes.

The larva has two peduncle compartments, IP and LP, as compared to the single compartment (PED) of the adult. The larval LP compartment with some of its MBINs becomes incorporated into the adult PED compartment, but the larval IP compartment is lost and not replaced. Its two MBINs (*Figures 3 and 5*) and at least two of its three MBONs (*Figure 4*, *Figure 6—figure supplement 3*) leave the larval MB and become parts of adult non-MB circuits. The IP compartment is unusual because of its isolation from the other MB microcircuits. The latter are highly interconnected by one- or two-step connections from the MBONs of one compartment to the MBIN of others. The IP MBINs, however, receive no MBON feedback from the other compartments (*Eschbach et al., 2020*). Likewise, the IP MBONs provide the least amount of crosstalk to the other larval compartments. Its isolation suggests that the IP is involved in a type of learning distinct from that handled by the rest of the larval MB. This may involve temperature-based learning since the greatest input from the small number of thermosensory Kenyon cells is onto MBON-b3 (*Eichler et al., 2017*). Since we found no corresponding compartment in the adult MB, the type of learning that IP mediates may be restricted to the larva.

The remaining seven larval compartments are incorporated into adult MB compartments, especially those containing γ neuron axons (*Figure 9A*). Their larval MBINs and MBONs typically also function in the mature MB, although the four PAM MBINs that project to the distal-most larval compartments (SHA, UT, IT, and LT) die at the end of the larval stage. The larval MBONs from these distal compartments, though, survive and innervate compartments at similar positions along the adult medial lobes, although some shift their dendrites from γ into β or β' compartments. For example, the larval SHA compartment corresponds to the adult γ3 compartment; of its two MBON-h's, one (MBON 08) innervates the adult γ3 compartment while the other (MBON 09) has dendrites in β'1 as well as γ3. The MBONs of the larval distal 'toes' (the UT, IT, and LT compartments) distribute their dendrites amongst the adult distal medial lobe compartments (γ4, γ5, β2, and β'2). The basal larval compartments, LP, LA, and LVL, correspond to adult compartments PED, γ1, and γ2, respectively. Most of their MBINs and MBONs stay within this set of compartments but there is some shuffling amongst them. The general pattern is that larval MBINs and MBONs that supply the seven larval compartments that are incorporated into the adult MB continue to function in the adult MB, while those that innervate the three larval-specific compartments retract from the MB and join non-MB circuits of the adult brain.

From the perspective of the mature, adult MB, 10 of its 16 compartments have axon cores from the postembryonic born α'β' or αβ Kenyon cells. The α and α' compartments are supplied almost exclusively by postembryonic-born MBINs and MBONs (*Figure 8*, *Table 2*). By contrast, the β and β' compartments show a mixed picture: their input is provided exclusively by postembryonic-born PAM neurons (*Figure 8A*; *Lee et al., 2020*), while their known outputs are through embryonic-born neurons from the larval MB (*Figure 4A*).

## Transmitter stability and shifts in compartments through metamorphosis

The input and output transmitters associated with MB compartments of the larval and adult systems are similar but not identical (*Figure 9B*). For the MBINs, the calyx receives octopaminergic input from the same neurons in the two stages. The adult calyx also receives a sparse serotonin input from a remodeled CSD neuron that projects from the contralateral antennal lobe (*Roy et al., 2007*). The compartments of the larval lobe system are primarily supplied by dopaminergic neurons, but the UVL and LVL compartments also have octopaminergic input (*Saumweber et al., 2018*). The octopamine input to the lobes is reduced in the adult MB, though, because OAN-e1/PPL1-SMP retracts completely from the larval UVL compartment and innervates the superior medial protocerebrum in the adult, and OAN-g1/OA-VPM3 reduces its MB input to a sparse innervation of the adult γ1 compartment (*Busch et al., 2009*).

The PPL1 and PAM clusters provide the major dopamine input to the lobes. The number of PPL1 neurons is about the same in the larval and adult stages, but we find that those innervating vertical lobe compartments differ in the two stages. In contrast, the PAM cluster innervation of the medial lobes is dramatically increased as the four neurons in the larva are replaced by about 150 neurons in the adult. Appetitive conditioning in the adult is complex with different medial lobe compartments receiving PAM input from different brain regions (*Li et al., 2020*) and providing reward information based on diverse factors such as sugar sweetness, nutritional value, and water (*Huetteroth et al., 2015*; *Lin et al., 2014*). Also, reproduction-related learning is also mediated through sets of PAM

neurons (e.g., the PAM 01 ( = γ5) neurons; *Zhao et al., 2018*). Even with only four PAM neurons, though, the larva shows selectivity in its reward learning (*Thum and Gerber, 2019*). For example, inhibition of DAN-h1 interferes with the positive association and odor with a fructose reward, but not with either amino acid or low salt rewards (*Saumweber et al., 2018*).

An adult-specific, modulatory input to the MB circuitry is provided by the paired, larval-born DPM neurons (*Mayseless et al., 2018*) that innervate all the compartments of the adult peduncle and lobes (but not the calyx). DPM neurons release serotonin (*Lee et al., 2011*) and peptides produced from the *amnesiac* gene (*Waddell et al., 2000*). These neurons are required for a delayed memory trace that appears in the MB about 30 min after training (*Yu et al., 2005*), and they participate in two forms of intermediate term memory, anesthesia-sensitive memory via the *amnesiac* gene, and anesthesia-resistant memory via serotonin and the *radish* gene (*Lee et al., 2011*). Anesthesia-resistant memory involving the *radish* gene also occurs in larvae (*Widmann et al., 2016*). As the DPM neurons are postembryonic, it is unknown whether larvae use another modulatory neuron for anesthesia-resistant memory.

On the MBON side, the output from the calyx in both larva and adult is cholinergic. In the larval stage, both MBON-a1 and -a2 receive similar input from Kenyon cells and from OAN-a1 and -a2; MBON-a1 also excites MBON-a2 via axo-axonic synapses (*Eichler et al., 2017*). MBON-a2/MBON 22 persists as the major cholinergic output from the adult calyx, while MBON-a1/MBON 29 shifts from the calyx to adult medial lobe compartments. MBON-c1/MBEN-CA is added to the adult calyx, but since we do not know its mature anatomy, we cannot speculate on its function.

For the vertical lobe system, the output transmitters from the distal compartments of the adult medial and vertical lobes largely conform to those from the corresponding regions of the larval medial and vertical lobes (*Figure 9B*). The adult α' and α (except α1) compartments have cholinergic output. The larval UVL compartment similarly has cholinergic output, but the transmitters of the MBONs from the IVL compartment are unknown.

The conservation of glutaminergic output of the distal medial lobe compartments between larva (UT, IT, and LT) and adult (γ4, γ5, β2, and β'2) comes from the same neurons being used in the two stages. The adult also has glutaminergic output from the α1 compartment provided by the two MBON 07s. These feedback on the α1 PAM neurons, thereby making a recurrent loop that is essential for appetitive long-term memory formation in the adult (*Ichinose et al., 2015*). We could not determine the developmental origin of MBON 07. In the larva, though, MBON-e2/MBON 03 provides an analogous glutaminergic output from the vertical lobes. It also feeds back onto its input neuron (OAN-e1) (*Eichler et al., 2017*), perhaps providing an analogous circuit to support long-term memory formation in the larva.

The compartments at the bases of the lobes and peduncle show some scrambling of neurotransmitter output through metamorphosis (*Figure 9B*). The output from the larval LP compartment is cholinergic while the corresponding adult PED compartment has GABAergic output. The opposite shift is seen in the LVL ( = γ2) compartment, which has GABAergic output in the larva but cholinergic output in the adult. As seen in *Figure 9C*, MBON-c1/MBEN-CA provides the cholinergic output from the larval LP compartment, but at metamorphosis it retracts from this compartment and dendrites from MBON-d1/MBON 11 invade the peduncle to provide the adult GABAergic output. The other compartmental shift in transmitter involves the larval LVL compartment; MBON-g1 and -g2 provide GABAergic output from this compartment, but at metamorphosis, they trans-differentiate to become central complex neurons. They are replaced by postembryonic-born MBON 12 (*Figure 8C*), which then provides cholinergic output for the adult γ2 and α'1 compartments. Thus, the compartmental shifts in output transmitters that occur at metamorphosis do not involve individual MBONs changing their transmitter. Rather, MBON recruitment, MBON loss, and MBONs shifting compartments combine to provide differences in transmitter landscapes in the two stages.

## Circuit-level implications of changes through metamorphosis

The examples of DAN-d1/PPL1 03 and DAN-c1/PPL1 01 depicted in *Figure 9C* show that shifting partners through metamorphosis can dramatically alter a neuron's function. In the larva, pairing of DAN-d1 stimulation with an odor induces short-term aversive conditioning, whereas a similar pairing with DAN-c1 does not (*Eschbach et al., 2020*; *Weiglein et al., 2021*). Their functions change in the adult, though, where DAN-c1/PPL1 01 becomes sufficient to induce short-term aversive conditioning

to a paired odor (*Aso et al., 2012*; *Das et al., 2014*), while DAN-d1/PPL1 03 becomes involved in higher levels of memory consolidation (*Owald et al., 2015*). The metamorphic changes in the functioning of these two MBINs comes from changing their MBON partners. In its larval form, DAN-d1/PPL1 03 works through MBON-d1/MBON 11 in establishing short-term aversive conditioning. This MBON functions similarly in the adult (*Aso et al., 2012*; *Perisse et al., 2016*), but its adult input is provided by DAN-c1/PPL1 01 (*Figure 9C*) rather than DAN-d1/PPL1 03. DAN-d1/PPL1 03, in turn, instructs a new partner, MBON 12, a cholinergic, postembryonic-born neuron that provides feedback excitation to DAN-d1/PPL1 03 and feeds across to a set of medial lobe MBONs whose activity promotes avoidance behavior, while their suppression promotes approach (*Owald et al., 2015*). These interactions provide a pathway in the adult to mediate memory re-consolidation and extinction (*Felsenberg et al., 2017*; *McCurdy et al., 2021*).

The adult form of DAN-c1/PPL1 01 has the added complexity that the type of learning it supports reverses depending on the time of its activity relative to the paired odor stimulus (*König et al., 2018*). Its activation after presentation of the odor reinforces avoidance of the odor, but if the odor is presented after DAN-c1/PPL1 01 terminates its activity. The fly then shows a 'relief' response and the odor becomes attractive (*König et al., 2018*). It does not have such a function in the larva (*Eschbach et al., 2020*). In the larva, a similar time-dependent switch from appetitive to aversive learning is mediated through DAN-f1 (*Weiglein et al., 2021*), a neuron that becomes incorporated into non-MB circuits in the adult (*Figure 3B*).

Hence, the persisting MB neurons rearrange their connections at metamorphosis as some neurons are lost from the MB via trans-differentiation (e.g., MBON-c1) and other, adult-specific neurons are added (e.g., MBON 12). Such changes likely reflect ad hoc solutions that enable the construction of a larval circuit without needed late-born cell types by using other early-born neurons that display

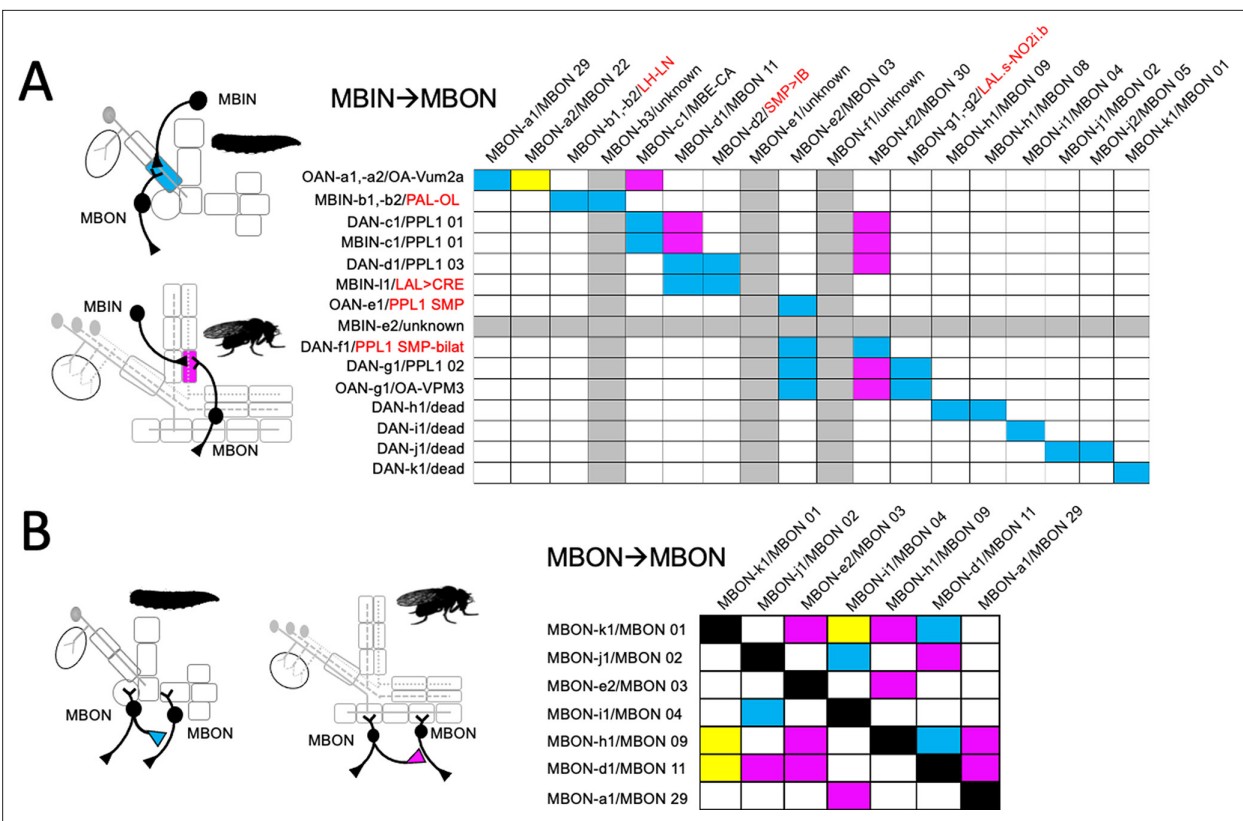

**Figure 10.** Fate of circuit connections in the mushroom body (MB) through metamorphosis. (**A**) Matrix showing the overlap of mushroom body input neuron (MBIN) axon terminals with mushroom body output neuron (MBON) dendrites in the same compartment. MBIN-MBON overlap only in the larval stage (blue), only in the adult stage (magenta), or in both stages (yellow). Rows and columns that are grayed out for cells whose identity is unknown for either the larval or adult stage. Larval/adult names are provided for each cell, with the red names being the terminal identity of neurons that do not innervate the adult MB. (**B**) Matrix showing the cells that make MBON-MBON connections only in their larval configuration (blue), only in their adult configuration (magenta), or in both configurations (yellow).

temporary phenotypes to make up for the missing cells. At metamorphosis, when the appropriate cell types are finally available, the temporary MB neurons trans-differentiate to evolutionarily ancestral phenotypes and the remaining neurons rewire into the adult circuit.

*Figure 10A* summarizes the connections between uni-compartmental MBINs and MBONs that are found in the larva (blue), the adult (red), or both (yellow). Stable MBIN-MBON pairing persists through metamorphosis only in the calyx neuropil, which contains the Kenyon cell dendrites. By contrast, the combination of compartment shuffling, trans-differentiation and neuronal death in the lobe compartments results in a lack of uni-compartmental MBIN to MBON pairings persisting from larva to adult.

Besides MBIN to MBON connections, the compartments of both the larva (*Eichler et al., 2017*; *Eschbach et al., 2020*) and the adult (*Aso et al., 2014*; *Li et al., 2020*) are highly interconnected, both by MBON-to-MBON connections and by feedback and feed-forward connections from MBONs back to MBINs. For MBON-to-MBON interactions, larval (*Eichler et al., 2017*) and adult (*Li et al., 2020*) connectivity data are available for seven of the MBONs that function in both circuits (*Figure 10B*). There are 42 possible pair-wise connections amongst these cells. These MBONs are more highly interconnected in their adult configuration compared to their larval one: their adult configuration shows 13 connections (31% of possible connections), while their larval configuration has only 7 (17%). Importantly, only three of these connections (7%) are present in both larva and adult. This percentage is similar to the 5% predicted if the two stages were wired up independently at their respective frequencies. This low level of shared connections suggests that in both their larval and the adult configurations, the MBONs interconnect in a way that is best adapted to the respective life stage.

Experiments on aversive conditioning of *Drosophila* larvae suggested that the memory from larval training can persist through metamorphosis (*Tully et al., 1994*). We find that within the MB lobe system, none of the MBIN-MBON pairings persist (*Figure 10A*) and persisting MBON-to-MBON connections are rare (*Figure 10B*). At this level, our anatomical findings do not identify any simple circuit elements that would support the persistence of an associative memory trace from larva to adult. Thus, a surviving memory trace would need to involve more complex anatomical pathways, such as described by *Eschbach et al., 2020*. However, this cannot be addressed in this study.

Our failure to find anatomical support in *Drosophila* for persistence of a memory trace from larva to adult should not be generalized to other insects with a larval stage. There is evidence that associative learning in moth caterpillars and beetle grubs can carry through to the adult (*Blackiston et al., 2015*; *Blackiston et al., 2008*). Larvae of butterflies and beetles have an extended embryonic development compared to *Drosophila,* and they hatch with a more complex larval nervous system. Consequently, more of the neuron types needed to make their MB are available to these embryos, likely making these insects less dependent on appropriating other neurons to temporarily function in the larval MB. A higher number of MB neurons persisting from the larva to adult increases the likelihood that a memory trace could persist from one stage to the other.

## Metamorphic changes of the larval neurons

Although there are examples of neurons that change their neuropeptides during postembryonic life (*Veverytsa and Allan, 2012*), our study did not find any neuron that changed its small molecule transmitter. The neurons did, though, show a great range of morphological changes. At one end of the spectrum were neurons like DAN-d1/PPL1 03 (*Figure 3B*) and MBON-j1/MBON 02 (*Figure 4B*), whose larval and adult forms are very similar. At the other end of the spectrum are MBIN-b/PAL-OL and MBON-g/LALs-NO$_i$.b (*Figure 4B*), which trans-differentiate into adult neurons that bear no similarity to their larval forms.

Neurons possessing the same form in both the larval and adult stages are like those of direct developing insects because they undergo their full developmental trajectory during embryogenesis and achieve their mature form at hatching. Other larval neurons have a morphology that appears to be based on pausing their developmental trajectory at an intermediate step and using this intermediate form as the basis for their larval morphology. The larval octopaminergic cells, OAN-a1 and -a2, fit this pattern. Their larval neurons stop at the MB calyx but in their adult form (OA-VUMa2), they extend beyond the calyx to form major arbors in the lateral horn neuropils (*Busch et al., 2009*). A similar strategy occurs for thalamic neurons in the developing mammalian visual system (*Kanold et al., 2003*). These early-born neurons arrive at the cortical subplate prior to the birth of their granule cell targets in the visual cortex. They assume an intermediate phenotype, synapsing with the subplate

neurons, but after the granule cells are born, they lose these temporary connections and grow into the cortex to find their final targets.

For other neurons, however, their larval form cannot be explained as a simple arrest along an ancestral developmental trajectory. The vertical axon branch of larval γ Kenyon cells is not seen as an intermediate stage in the development of γ neurons of direct developing insects such as crickets (*Malaterre et al., 2002*). The larval form of these neurons requires a developmental deviation that adds new features to adapt the neurons' morphology to the larval stage. The extreme version of adding larval-specific novelty is the radical change in cellular phenotype seen in trans-differentiating neurons like MBON-g and MBIN-b.

Cells that undergo trans-differentiation, like MBON-g/LALs-NO$_2$i.b, show extensive pruning at the start of metamorphosis. Some neurons that have essentially the same morphology in larva and adult, like MBON-j1/MBON 04 and APL (*Figure 6—figure supplement 1*), also show extreme pruning. But others, like DAN-c1/PPL 01 or DAN-d1/PPL1 03 (*Figure 5*), show only moderate arbor loss. This variation reflects the fact that while pruning is due to a cell autonomous developmental program triggered by the steroid ecdysone (*Lee et al., 2000*; *Schubiger et al., 1998*), its trajectory may be guided in some cells by local interactions with pre- or postsynaptic targets. The importance of local interactions was experimentally examined for the pruning of APL (*Mayseless et al., 2018*). Blocking ecdysone action in APL inhibits its pruning response. The selective inhibition of ecdysone action in γ Kenyon cells, the main synaptic partners of APL, similarly inhibits γ cell pruning but also that of the untreated APL. Thus, while steroid signaling is needed to activate the neuron's pruning program, the extent of neurite loss may depend on changes in synaptic partners. Similarly, for the MBINs, the larval LP and LA compartments retain γ neuron axons during pruning (*Watts et al., 2003*), and we see that their MBINs (DAN-c1 and DAN-d1) maintain most of their axonal tufts through the pruning period. By contrast, neurons of distal medial lobe compartments, which lose their γ neuron axons, prune extensively even though they grow back to a similar adult morphology (e.g., MBON j1/MBON 02; *Figure 6—figure supplement 1*).

## Relationship of lineage to metamorphic fates of MB neurons

The MBINs and MBONs of the adult CNS are produced by 10–15% of the ~100 NBs that construct each brain hemisphere. Most of the neuron types that serve as temporary larval MB neurons are recruited from these same lineages. Although larval NBs (*Sprecher et al., 2007*) and adult lineages (*Yu et al., 2013*) have been mapped and described, the two maps have not been reconciled. Indeed, in most cases, we do not know exactly which embryonic brain NB produces which postembryonic lineage. *Figure 11* presents the major embryonic and postembryonic lineages that produce MBINs and MBONs. The Kenyon cells are produced by the four MBps that begin dividing at mid-embryogenesis and only finish just before the emergence of the adult. Their earliest embryonic cells differ (*Kunz et al., 2012*), but the four NBs produce identical lineages after they begin Kenyon cell production. All the other neuroblasts make a small number of neurons embryonically, but then arrest and subsequently resume cycling late in the first larval instar. Their small size during the dormant period makes them difficult to track through this transition.

Most MBINs come from the PPL1 and the PAM clusters. We find that the generation of the adult PPL1 is split, with some members born in the embryo and initially functioning as larval MBINs, while others are born after neurogenesis resumes in the larva (*Figure 11*). The embryonic neuroblast that makes these neurons is CPd2 or 3 (*Saumweber et al., 2018*), and it appears to arrest in the midst of producing the PPL1 MBINs, a conclusion based on the observation that MBIN-c1 is born so late in embryogenesis that it is not yet incorporated into the MB circuitry at hatching (*Eichler et al., 2017*). The neuroblast is called DL2 when it reactivates in the larva and shows a type II pattern of division (*Boone and Doe, 2008*; *Walsh and Doe, 2017*). The first neurons that DL2 produces after it resumes dividing are the remaining PPL1 MBINs (*Ren et al., 2016*). The adult PPL1 neurons, therefore, appear to arise as a set of neurons that straddle the temporary arrest of the DL2 neuroblast. Those born in the embryo then function as MBINs in both the larva and the adult, while those born in the larva delay their maturation into MBINs until metamorphosis. Based on their clustering with the 'permanent MBINs,' the embryonic-born PPL1 neurons that serve as temporary larval MBINs (OAN-e1, MBIN-e2, and DAN-f1) arise in the same lineage but must be produced earlier in embryogenesis.

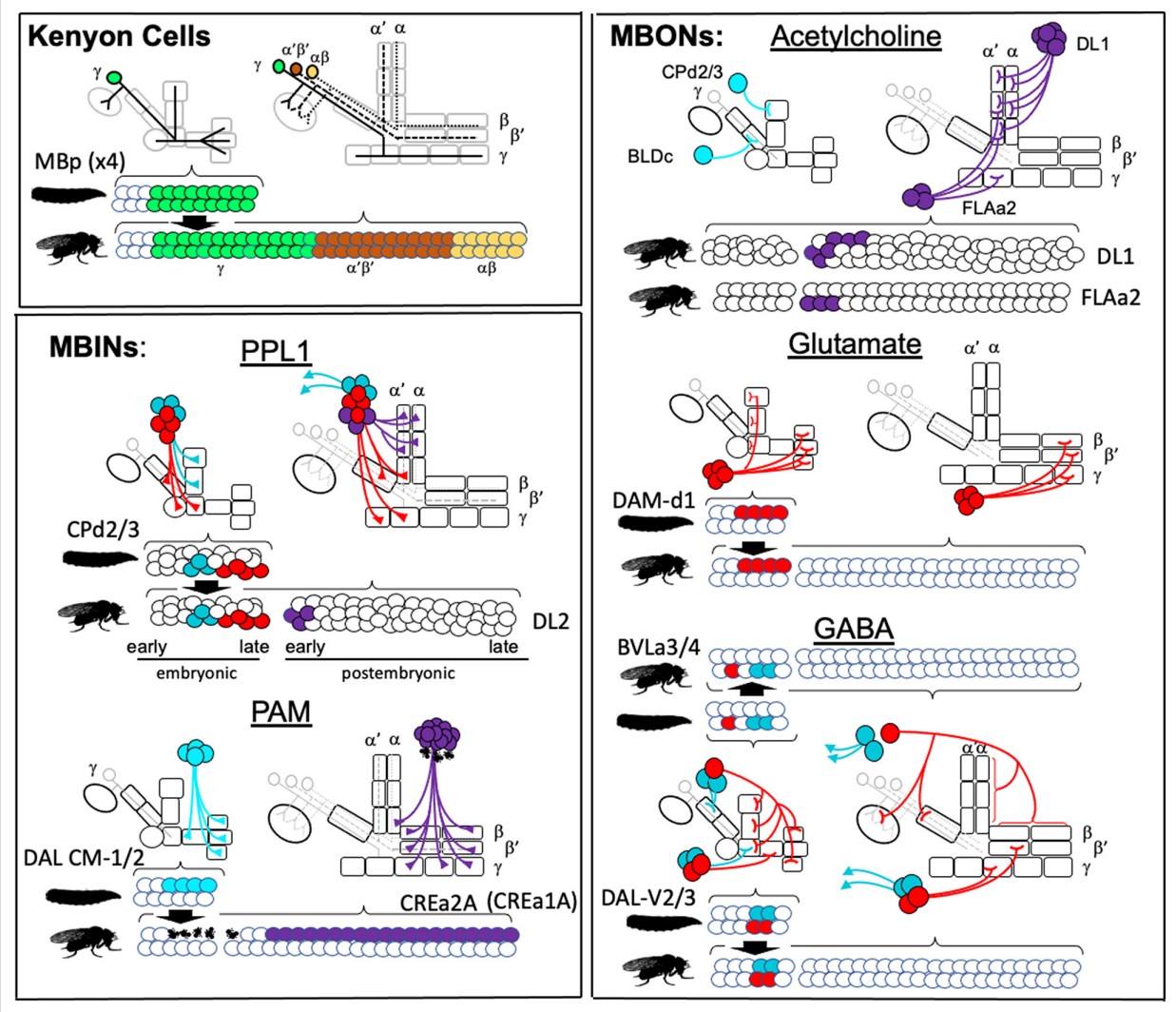

**Figure 11.** The lineage relationships of the major neuron types of the larval and adult mushroom bodies (MBs). The four Kenyon cell neuroblasts (MBp) divide continuously from mid-embryogenesis until just before adult emergence. Three classes of Kenyon cells (different colors) are made in succession. The remaining neuroblasts have discrete embryonic and postembryonic phases of neurogenesis; neurons made during the first phase make the larval CNS. At metamorphosis, they are combined with neurons from the postembryonic phase to make the adult CNS. Origins and fates of MB neurons: (red) neurons that function in both the larval and adult MB; (blue) neurons that function in the larval MB but switch to non-MB circuits in the adult; (purple) postembryonic-born cells that function only in the adult MB: (white) neurons that function outside of the MBs. The name of the larval neuroblast and its embryonic lineage is paired with the most likely postembryonic lineage. Most postembryonic lineages are type I, in which progeny arise pairwise fashion from division of successive ganglion mother cells (shown as paired lines of cells). DL1 and DL2 show a type II pattern of division that increases the number of neurons produced because each neuroblast division produces an intermediate precursor cell that then divides to produce a small number of ganglion mother cells (shown as more disorganized clusters). See text for details.

The adult PAM neurons are all born during the postembryonic period and are produced by two closely related lineages, CREa1A and CREa2A (*Lee et al., 2020*). The first few postembryonic neurons born in both lineages are not PAM neurons. However, both neuroblasts soon switch into a repetitive mode, in which the 'Notch-ON' daughter (see below) of each GMC becomes a PAM neuron, a pattern repeated through the next ~75 GMCs. The generation of this large neuronal class at the end of their lineages is consistent with the general pattern that neurons within a hemilineage become more similar as a neuroblast progresses deeper into its lineage (*Lee et al., 2020*).

Embryonic DAL CM1 and 2 are likely the embryonic version of CREa1A and 2A. One or the other makes four PAM neurons for use in the larva, but these subsequently die at metamorphosis. These temporary PAM neurons also seem to be born at the end of the embryonic neurogenic phase, because like MBIN-c1 described above, one of them, DAN-h1, is not yet incorporated in the MB circuitry at

hatching (*Eichler et al., 2017*). Interestingly, after the CREa2A neuroblast resumes dividing in the larva, its first Notch-ON daughter degenerates right after its birth (*Lee et al., 2020*). We suggest that towards the end of its embryonic phase, the CREa2A neuroblast produces a set of GMCs, whose Notch-OFF daughter is required but whose Notch-ON daughter is 'unneeded' and fated to die. This pattern is carried through into the start of the postembryonic phase, as evident by the first postembryonic, Notch-ON daughter dying immediately after its birth. The Notch-ON daughters produced during embryogenesis, though, defer their deaths until metamorphosis and serve as temporary PAM neurons for the larva.

Members of various groups of MBONs are also related by lineage and by transmitter type. The adult has eight types of cholinergic MBONs that provide output from the α and α' compartments (*Aso et al., 2014*). Three come from the FLAa2 neuroblast and are produced at the beginning of its postembryonic phase (*Figure 8C*). The remainder are produced within the postembryonic period by the DL1 neuroblast, another neuroblast that shows a type II pattern of division. Its divisions result in a series of GMCs that each divide to produce a cholinergic MBON and central complex neuron (*Figure 8D*). Neither DL1 nor FLAa2 appear to contribute embryonic-born cells to the larval vertical lobe. We do not know the transmitters of the two IVL MBONs, but MBON-e1, from the UVL compartment, is cholinergic (*Eichler et al., 2017*). Curiously, it comes from the CPd2/3 group, a group that includes the PPL1 MBINs.

Besides the multicompartmental neuron, APL, the adult has four types of GABAergic MBONs (*Aso et al., 2014*). The β'1 compartment is innervated by eight MBON 10-type neurons. We do not know their origin although the large number of neurons in this group suggests that they are born during the postembryonic period. The remaining three types of GABAergic MBONs are embryonic-born and wholly or partially associated with γ compartments in both the larva and the adult. MBON-d1 comes from the DAL CM-1/2 group (*Saumweber et al., 2018*), the same group responsible for the PAM neurons. Of special interest, though, are MBONs 08 and 09 from the DAL-V2/3 group. These neurons have overlapping compartmental functions in the adult, but are identical in the larva, showing essentially the same synaptic connectivity within the MB (*Eichler et al., 2017*). This lineage produces four, rather than two, GABAergic MBONs in the larva, but the additional two larval cells (MBON-g1 and -g2) trans-differentiate into adult central complex neurons. The BLVa3/4 group that produces APL also produces two larval-specific GABAergic neurons, MBON-b1 and -b2, that become lateral horn neurons in the adult (*Figure 4*, *Table 3*).

The adult medial lobe compartments are supplied by seven types of glutamatergic MBONs (*Aso et al., 2014*). We do not know the origin of the adult MBON 07, but the six other types are embryonic born, with the majority coming from the DAM-d1 lineage (*Saumweber et al., 2018*). These six types provide sufficient glutamatergic MBONs to cover larval medial lobe function, with an additional type leftover, MBON-e2/MBON 03, which is shifted to the larval vertical lobe. It provides glutamatergic output from the larval vertical lobe, perhaps analogous to that provided by MBON 07 from the adult α1 compartment.

As illustrated in *Figure 12A*, neuronal identity is established within a lineage by relative birth order of the GMCs (*Kohwi and Doe, 2013*; *Doe, 2017*; *Miyares and Lee, 2019*) and by symmetry-breaking to establish different fates of the two daughters of the GMC division. Relative birth order is encoded at the start of neurogenesis by the sequential expression of a series of transcription factors, *Hunchback → kruppel → pdm → castor*, in the neuroblast as it divides. The transcription factor expressed at the time of division is passed into the GMC and then into her two daughter cells, thereby providing a record of relative birth order. In *Drosophila*, an embryonic neuroblast typically reaches *castor* expression by the time of its arrest late in embryogenesis (*Isshiki et al., 2001*) and it typically resumes expressing *castor* when it reactivates in the larva (*Maurange et al., 2008*). The phenotypes of the two siblings arising from the GMC division are established through Notch signaling, which results in a Notch-ON ('A') fate and a Notch-OFF ('B') fate (*Skeath and Doe, 1998*). Successive cells sharing that same Notch state typically have similar properties, resulting in the neuroblast producing two hemilineages (A and B) of neurons of related form and function (*Truman et al., 2010*; *Mark et al., 2021*). The information on relative birth order and hemilineage status then acts through a set of terminal selector genes (*Hobert and Kratsios, 2019*) to produce a characteristic neuronal phenotype. *Figure 12B and C* speculates on how these two mechanisms might be exploited to alter a neuron's phenotype for specialized use in the larva.

**Table 3.** Comparison of transmitter expression in larval and adult forms of mushroom body output neurons (MBONs) and mushroom body input neurons (MBINs).

| Neuron: larva/adult | Larval transmitter | Adult transmitter | Larval ref | Adult ref |
|---|---|---|---|---|
| **MBINs** | | | | |
| OAN-a1,a2/OA-VUM2a | Octopamine | Octopamine | Selcho et al., 2014 | Busch et al., 2009 |
| MBIN-b1,b2/PAL-OL | TH-positive | TH-positive | Eichler et al., 2017 | Mao and Davis, 2009 |
| DAN-c1/PPL1 01 | Dopamine | Dopamine | Selcho et al., 2009; Eichler et al., 2017 | Aso et al., 2014; Li et al., 2020 |
| MBIN-c1/PPL1 01 | TH-positive | TH-positive | Eichler et al., 2017 | Aso et al., 2014; Li et al., 2020 |
| DAN-d1/PPL1 03 | Dopamine | Dopamine | Selcho et al., 2009; Eichler et al., 2017 | Aso et al., 2014; Li et al., 2020 |
| MBIN-ll/TD_LAL >cre | TH-positive | Unknown | Eichler et al., 2017 | |
| OAN-e1/PPL1-SMP | Octopamine | TH-positive | Eichler et al., 2017 | Mao and Davis, 2009 |
| DAN-f1/PPL1-bilat | Dopamine | TH-positive | Selcho et al., 2009; Eichler et al., 2017 | Mao and Davis, 2009 |
| DAN-g1/PPL1 02 | Dopamine | Dopamine | Selcho et al., 2009; Eichler et al., 2017 | Aso et al., 2014; Li et al., 2020 |
| OAN-g1/OA-VPM3 | Octopamine | Octopamine | Selcho et al., 2014; Eichler et al., 2017 | Busch et al., 2009 |
| **MBONs:** | | | | |
| MBON-a1/MBON 29 | Acetylcholine | Putative ACh | Eichler et al., 2017 | Li et al., 2020 |
| MBON-a2/MBON 22 | Acetylcholine | Putative ACh | Eichler et al., 2017 | Li et al., 2020 |
| MBON-b1,b2/LH-LN | GABA | Putative GABA | | Dolan et al., 2019 |
| MBON-c1/MBE-Calyx | Acetylcholine | Unknown | Eichler et al., 2017 | |
| MBON-d1/MBON 11 | GABA | GABA | | Aso et al., 2014; Li et al., 2020 |
| MBON-d2/SMP>IB | Unknown | Unknown | | |
| MBON-e2/MBON 03 | Glutamate | Glutamate | Eichler et al., 2017 | Aso et al., 2014; Li et al., 2020 |

*Table 3 continued on next page*

*Table 3 continued*

| Neuron: larva/adult | Larval transmitter | Adult transmitter | Larval ref | Adult ref |
|---|---|---|---|---|
| MBON-f2/MBON 30 | Unknown | Putative glutamate | | *Li et al., 2020* |
| MBON-g1,g2/LAL.s-NO2i.b | GABA | Unknown | *Eichler et al, 2017* | |
| MBON-h1/MBON 09 | GABA | GABA | *Eichler et al., 2017* | *Aso et al., 2014; Li et al., 2020* |
| MBON-h2/MBON 08 | GABA | GABA | *Eichler et al., 2017* | |
| MBON-i1/MBON 04 | Glutamate | Glutamate | *Eichler et al., 2017* | *Aso et al., 2014; Li et al., 2020* |
| MBON-j1/MBON 02 | Glutamate | Glutamate | *Eichler et al., 2017* | *Aso et al., 2014; Li et al., 2020* |
| MBON-j2/MBON 05 | Glutamate | Glutamate | *Saumweber et al., 2018* | *Aso et al., 2014; Li et al., 2020* |
| MBON-k1/MBON 01 | Glutamate | Glutamate | *Eichler et al., 2017* | *Aso et al., 2014; Li et al., 2020* |
| APL/MB-APL | GABA | GABA | | *Aso et al., 2014; Li et al., 2020* |

TH, tyrosine hydroxylase.

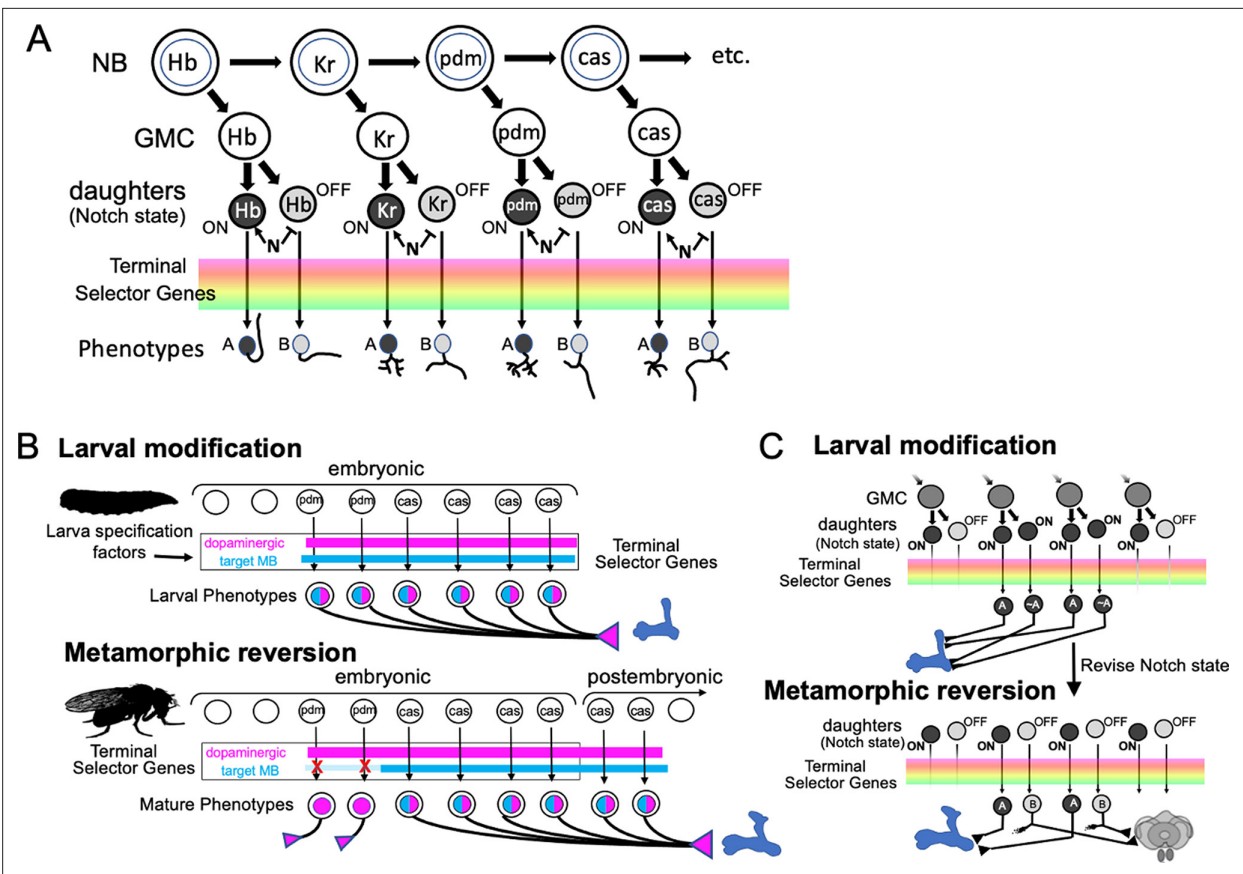

**Figure 12.** The origin of spatial-temporal information used to determine neuronal phenotypes. (**A**) The early neuronal phenotypes within a lineage are determined by birth order of the ganglion mother cells (GMC). Birth order is encoded by a temporal program of transcription factor expression in the parent neuroblast (NB) as it divides. Transcription factor expression at the time of division is inherited by the GMC and its daughter neurons. Differences between the daughters is established by Notch (N) signaling with one sibling expressing the Notch-on ('A') fate and the other the Notch-off ('B') fate. This information, along with lineage identity factors, acts through a battery of terminal selector genes to establish neuronal phenotypes. (**B**) A hypothetical scheme to explain the metamorphic pattern of mushroom body input neuron (MBIN) recruitment and loss in the PPL1 cluster of dopamine neurons. It proposes that the Castor (cas) expressing neurons that are born in the DL1 lineage just before and after the embryonic neurogenic arrest are fated to become MBINs. The earlier born Pdm expressing neurons are also dopaminergic but their adult function is outside of the MB. Larval specification factors, though, modify how they interact with the terminal selector genes thereby transforming them into MBINs while the larval stage is maintained. (**C**) A hypothetical scheme using neurons of the DAL-V2/3 lineage to illustrate how sibling fates might be temporarily altered to recruit larval MBONs. In this scheme, two successive GMCs divide to produce one daughter that is an MBON and one that innervates the central complex, a dichotomy established by Notch signaling. With the evolution of the larva, Notch signaling is suppressed in the daughters during embryogenesis, allowing both to assume a similar fate – that of a larval MBON. With the reestablishment of normal Notch signaling at metamorphosis, the transformed daughter loses her MBON features and becomes a central complex neuron.

Information on birth order may underlie the changes seen in the PPL1 cluster (*Figure 12B*). As described above, production of the set of neurons that become adult PPL1 MBINs spans the period of neuroblast arrest. We speculate that this block of neurons expresses *cas*. In this scenario, their earlier born siblings that serve as temporary larval MBINs would likely express the previous gene in the series of temporal transcription factors, *pdm*. At metamorphosis the Pdm+ neurons remain aminergic (*Mao and Davis, 2009*) but shift to targets outside of the core MB circuitry. The speculative scheme depicted in *Figure 12B* proposes that in the adult, Cas and Pdm work via overlapping sets of terminal selector genes: both establish a dopaminergic phenotype, but only Cas activates genes targeting the neuron to the MB. The intercalation of the larval stage, though, would involve an additional 'larval specification factor,' whose presence alters the actions of Cas and Pdm on the terminal selector genes. In the presence of this hypothetical factor, Pdm also supports targeting to the MB, thereby transforming the Pdm+ group into MBINs, which innervate the larval vertical lobe. The subsequent loss of

the larval specification factor at metamorphosis results in the Pdm+ neurons' withdrawal from the MB and their redirection to non-MB targets.

Another way of recruiting neurons to the larval MB could involve modification of Notch signaling during a GMC division (*Figure 12C*). Our data on the DL1 lineage show six examples of GMCs whose division results in one daughter being an MBON and the other being a central complex cell (*Figure 8D*). We do not know the actual relationship of MBON-g/LAL.s-NO$_2$i.b to MBON-h/MBON 09 within DAL-v2/3 lineage, but we propose a relationship like that seen in the DL1 lineage, with one daughter becoming an MBON (08 or 09) and the other a central complex neuron (LAL.s-NO$_2$i.b). In this scenario, the four neurons come from two successive, embryonic-born GMCs. In the larva, however, both siblings become MBONs. This larval similarity could arise from altering embryonic Notch signaling. Typically, the 'Notch-ON' phenotype of the A sibling is established through the Notch target *Hey* (Hairy/enhancer-of-split like with a Y), a bHLH-O transcription factor (*Monastirioti et al., 2010*). Interestingly, in the Kenyon cell lineages, *Hey* expression is independent of *Notch* (*Monastirioti et al., 2010*) making both siblings 'Notch-ON' and identical. *Figure 12C* suggests that a similar change in Notch state may have occurred for relevant GMCs of the DAL-v2/3 lineage, thereby allowing both siblings to express the MBON fate in the larva. The reestablishment of the normal Notch relationship at metamorphosis might then cause the MBON-gs to lose their MBON characteristic and to acquire their appropriate phenotypes as central complex neurons.

## The generation of the larval mushroom body

The neuronal identity in the insect CNS is generated according to a spatial and temporal pattern that is highly conserved through evolution (*Thomas et al., 1984*). The earliest born neurons are often diverse sets of projection neurons that are the basis for the stereotyped tracts and commissures that characterize insect neuropils. Local interneurons are born later in lineages, and these become more similar as the lineage progresses (*Lee et al., 2020*). The evolution of the holometabolous larva involved a shortening of embryonic development, producing a simplified larval body form that could successfully compete for ephemeral food sources. This shortening of insect embryogenesis, though, had a profound impact on their neuroblast-based mode of neurogenesis, resulting in a neurogenic arrest before lineages were complete. Hence, not only do larvae hatch with fewer neurons than found in the mature nervous system, but they should have only the types of neurons characteristic of the early portion of each lineage. Such a truncation may produce mismatches between the neuron types that are needed and those that have been made, i.e., required late-born cell types may be missing

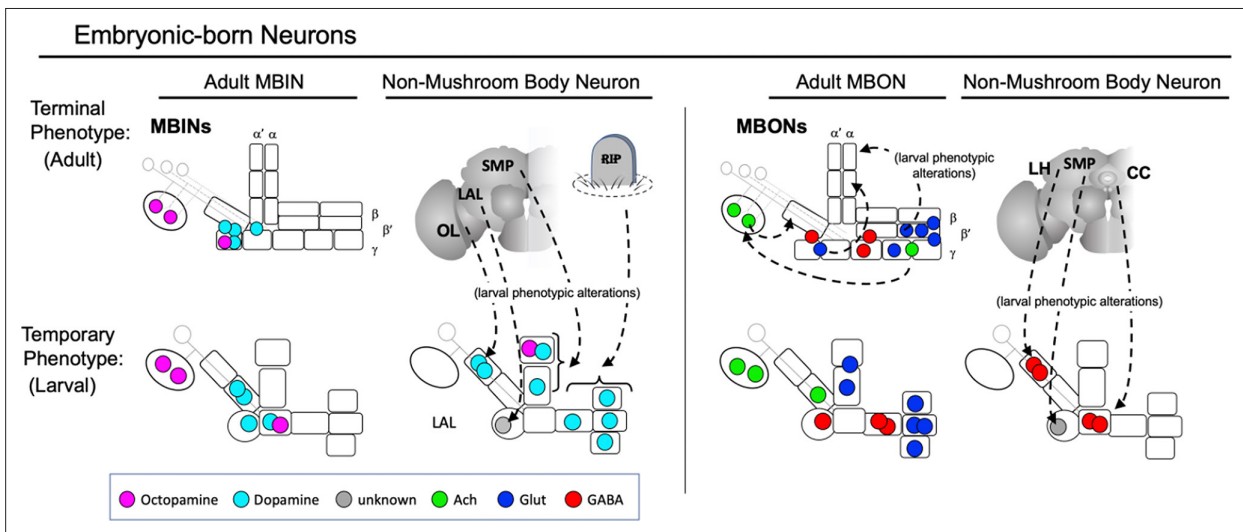

**Figure 13.** Summary of the relationship of the function of embryonic-born adult neurons to their temporary functions in the larval mushroom body (MB). Neurons that function in the MB of both larva and adults typically have similar positions in both structures, although some of the adult medial lobe mushroom body output neurons (MBONs) shift to the larval vertical lobes. Other neurons innervating the larval vertical lobes and the larval-specific intermediate peduncle compartment are fated for adult functions outside of the MB. CC: central complex; LAL: lateral accessory lobe; LH: lateral horn; OL: optic lobes; RIP: dead neurons; SMP: superior medial protocerebrum.

and the larva may have early-born cell types that it does not need! Our analysis of the developmental relationship of the larval to the adult MB provides insight into how these mismatches are resolved.

The relationships of Kenyon cell classes to the MB vertical and medial lobes are highly conserved (*Farris, 2005*; *Farris and Sinakevitch, 2003*). The γ Kenyon cells are born first, and their axons form a medial lobe. They are followed by branched, αβ-type neurons, whose vertical axon branch forms a vertical (α) lobe and whose medial, β branch joins the γ cell axons in the medial lobe. In direct developing insects like crickets (*Malaterre et al., 2002*), both γ and αβ-type Kenyon cells arise during embryogenesis to form the MB of the hatchling cricket, but the hatchling *Drosophila* has only the early-born γ neurons in its MB. In the absence of late-born αβ-type neurons, the γ neurons are modified with a larval-specific vertical axon branch that provides the basis for the larval vertical lobe. We find that the MBINs and MBONs that provide the input-output circuitry for the larval lobes are neurons that either function in the MB circuit in both larva and adult or function as MB neurons only temporarily in the larva. For the latter cells, we have concluded that their larval phenotype is a transient adaptation for the larval stage and their adult phenotype is more like that seen in their direct-developing ancestors. The deviation of a neuron's larval phenotype from its adult phenotype therefore represents an adaptation to accommodate the lack of needed, late-born neurons.

*Figure 13* summarizes how the derived phenotypes of the larval MB neurons relate to their mature phenotypes in the adult brain. Previous studies on the abdominal nervous system showed that many larval neurons come from a pool of neurons that die during embryogenesis in direct developing insects, but their death is delayed in metamorphic insects to allow these neurons to function in the larval CNS (*Truman, 2005*). In the MB circuit, we found that the only neurons that were recruited from this pool of 'doomed' neurons were the four larval PAM neurons. The death, though, is perplexing because they are a neuron type needed by the adult, as shown by the addition of ~150 postembryonic born PAM neurons to the adult brain. We think that the death of the larval PAM neurons may be related to mechanisms that allow a particular neuron class to be expanded within its lineage. Two clusters of dopamine neurons homologous to the *Drosophila* PAM clusters are already enlarged in locusts (*Wendt and Homberg, 1992*), so this expansion likely occurred before the evolution of complete metamorphosis. Possibly, very ancient insects made only a few PAM-like neurons, and these appeared early in their lineage at an appropriate timing for innervating the early-born γ neurons. If different neuron types maintain their relative order of production within this lineage, then a major expansion of an early-born PAM class would greatly delay the production of later neuron types. A modification that resulted in a PAM phenotype reappearing at the end of the lineage, however, would provide a late-born PAM class that could increase in number without interfering with the appearance of the earlier neurons. Developmental complications of integrating two sets of PAM cells (a small early set and an expanded late set) may have favored the late-born neurons, and thereby necessitated programmed cell death to remove the early-born PAM cells. This scenario brings up the intriguing possibility that after being consigned to the graveyard of unneeded embryonic cells for millions of years, the evolution of the larval stage provided a reason for these neurons to be 'resurrected' as temporary early-born PAM cells for use in the larval CNS.

Except for the PAM neurons, embryonic-born neurons that function in the larval MB also function in the adult, either in its MB or in non-MB circuits (*Figure 13*). The MBINs and most of the MBONs whose terminal role continues in the MB have similar positions in both the larval and the adult MB. For a few MBONs, though, their larval MB function differs markedly from their adult MB function. The embryonic born, glutaminergic MBONs provide informative examples. The increased number of medial lobe compartments in the adult are innervated by more glutaminergic MBONs than are the smaller number of larval medial lobe compartments. We find that the full set of adult glutaminergic MBONs are made during embryogenesis, however. Most take up similar positions in the larval medial lobe while the 'extra' cells are modified to function in the larval vertical lobe. In this way, they temporarily substitute for the late-born cells that normally supply vertical lobe compartments. Many of the neurons that supply the larval-specific compartments of the intermediate peduncle and vertical lobes are neurons fated for non-MB circuits, and modified in the larva for only temporary use in the MB. Many are fated for adult-specific neuropils, such as the central complex or the optic lobes (*Figure 13*). These temporary, larval MB neurons appear most frequently in lineages that also make permanent MBONs or MBINs. These cells already have the appropriate lineage information to

produce the desired phenotypes; they only need to alter the phenotypic read-out of their temporal information and/or Notch state.

We suggest that a hypothetical 'larval specifying factor(s)' is involved in altering the interpretation of the temporal and spatial factors that establish neuronal phenotypes. The expression of such a factor would maintain the larval state but its disappearance at metamorphosis would mean that the larval phenotype could no longer be maintained, and the neurons could change to a mature phenotype appropriate to their temporal and spatial instructions. Indeed, recent studies show that the BTB/POZ transcription factor, Mamo, appears at metamorphosis and is need for the partially dedifferentiated larval neurons to acquire their adult state (*Lai et al., 2022*). While the existence of a larval specification factor is hypothetical in this context, three transcription factors, *chinmo*, *broad*, and *E93,* act as master genes to specify the different life stages of *Drosophila* (*Truman and Riddiford, 2022*). Also, in some insects, the larval stage is actively maintained by the sesquiterpene hormone, juvenile hormone,

**Table 4.** Split GAL4 lines used to determine fates of larval mushroom body output neurons (MBONs) and mushroom body input neuron (MBINs).

| Cell name | Split line | Split line | Split line |
|---|---|---|---|
| MBIN-b1,b2 | **SS21716 [FS, TL)** | | |
| DAN-c1 | **SS03066 (FS, TL)** | **MB586B (FS)** | SS01702 (FS) |
| DAN-d1 | **MB328B (FS, TL)** | | |
| MBIN-l1 | **SS04484 (FS, TL)** | SS01624 (FS) | |
| OAN-e1 | **SS36923 (FS)** | SS01958 (FS) | |
| DAN-f1 (+DAN-c1) | **MB065b (FS, TL)** | MB145 (TL) | |
| DAN-g1 | **SS01716 (FS, TL)** | SS01755 (FS) | |
| OAN-g1 (sVPMmx) | **SS25844 (FS)** | **SS04268 (FS)** | |
| DAN-h1 | **SS01949 (NC,TL)** | SS01696 (NC) | MB440B (NC) |
| DAN-i1 | **SS01949 (NC,TL)** | MB196C (NC) | |
| DAN-j1 | **SS01949 (NC,TL)** | MB316B (NC) | MB340C (NC) |
| DAN-k1 | **SS01757 (NC,TL)** | MB198B (NC) | SS00616 (NC) |
| MBON-a1 | **SS00867 (FS,TL)** | **SS01417 (FS)** | |
| MBON-a2 | SS02006 (FS) | | |
| MBON-b1,b2 | **SS01708 (FS, TL)** | **SS04112 (FS)** | SS01959 (FS) |
| MBON-c1 | SS21789 (FS, TL) | | |
| MBON-d1 | **SS01705 (FS,TL)** | | |
| MBON-d2 | SS04231 (FS) | | |
| MBON-e2 | SS04559 (FS, TL) | | |
| MBON-f2 | **SS04328 (FS, TL)** | SS36248 (FS) | |
| MBON-g1,g2 | **SS02130 (FS, TL)** | SS02121 (FS) | |
| MBON-h1,h2 | **SS01725 (FS, TL)** | | |
| MBON-i1 | SS01771 | | |
| MBON-j1 | **SS01973 (FS,TL)** | SS01972 (FS) | |
| MBON-j2 | **SS00860 (FS,TL)** | | |
| MBON-k1 | **SS01962 (FS)** | **SS01980 (TL)** | |
| APL | **SS01671 (FS, TL)** | | |

Bold lines are the best lines for each cell.

FS, flip-switch immortalization; NC, no adult counterpart; TL, developmental timeline.

acting through its major target, the Krüppel-homolog 1 transcription factor (*Jindra et al., 2013*). These genes may provide an entry into discovering how the development of terminal fates can be temporarily diverted to produce a novel, larval identity.

## Materials and methods

### Fly stocks

*Drosophila* stocks were raised on standard corn meal molasses at either 25°C or room temperature. The genetic stocks used in this study are summarized in *Tables 4 and 5*. We initially examined both males and females for sexually dimorphic adult phenotypes. We found an obvious sexual dimorphism only for the MBIN-b1 and -b2 neurons. Beyond these neurons, then, we did not discriminate as to the sex of the animals that we used.

### Flip-switch treatments

The expression pattern seen in the late third- instar in stable spilt lines was maintained through metamorphosis using the flip-switch method described in *Harris et al., 2015*. Using a similar strategy of the gene-switch method (*Roman et al., 2001*), flippase was fused to the ligand-binding domain of the human progesterone receptor, rendering it dependent on progesterone or a progesterone mimic to move into the nucleus. Stable split lines were crossed to *pJFRC48-13XLexAop2-IVS-myrtdTomato in su(Hw)attP8; Actin5Cp4.6>dsFRT>LexAp65 in su(Hw)attP5; pJFRC108-20XUAS-IVS-hPRFlp-p10 in VK00005/TM6*.

We used the progesterone mimic mifepristone (RU486, Sigma-Aldrich; #M8046) to cause translocation of the flippase to the nucleus where it could then flip-out the STOP cassette in the Actin-LexAp65 transgene. We used surface application of RU486 to food vials. Parents were allowed to lay eggs in a food vial for a few days, then transferred to a fresh vial. Approximately 4 days after transfer, 60 µl of an ~10 mM RU486 stock solution (10 mg RU486 dissolved in 2 ml 95% ethanol) was applied to the surface of the food. At 24 hr after treatment, any larvae that had wandered and/or pupariated were discarded to ensure that test animals had fed on RU486 for at least 24 hr. At 48 hr after treatment, the subsequent wandering larvae and pupae (which had all fed on RU486 for 24–48 hr during the L3 stage) were collected and transferred to an untreated food vial. These animals were then dissected in Schneider's S2 culture medium as adults. This treatment results in constitutive LexA expression in any cells that express GAL4 during the L3 stage, but, because the RU486 persists at least partway through metamorphosis, neurons that start expressing in early to mid-metamorphosis also show up.

### Lineage-targeted twin-spot MARCM

Specific neuronal lineages were targeted using lineage-restricted drivers (*Awasaki et al., 2014*) to label sister clones with twin-spot MARCM (*Yu et al., 2009*). The Vnd-GAL4 driver permits targeting 18 fly central brain lineages, including the FLAa2 lineage (*Lee et al., 2020*); and stg14-GAL4 driver covers eight type II neuronal lineages, including the DL1 lineage (*Wang et al., 2014*). Twin-spot clones were induced at specific times after larval hatching and examined at the adult stage by immunostaining and confocal imagining, following published work (e.g., *Yu et al., 2010*).

### Preparation and examination of tissues

Tissues were dissected in phosphate-buffered saline (PBS, pH 7.8) and fixed in 4% buffered formaldehyde overnight at 4°C. Fixed tissues were rinsed in PBS-TX (PBS with 1% Triton X-100, Sigma), then incubated overnight at 4°C in a cocktail of 10% normal donkey serum (Jackson ImmunoResearch), 1:1000 rabbit anti-GFP (Jackson ImmunoResearch), 1:40 rat anti-N-Cadherin (Developmental Studies Hybridoma Bank), and 1:40 mouse anti-Neuroglian or a 1:200 dilution of mouse anti-FasII (both Developmental Studies Hybridoma Bank). For visualization of tdTomato, a 1:500 dilution of rabbit anti-DsRed (ClonTech) was substituted for the anti-GFP and the anti-Neuroglian was omitted. After repeated rinses with PBS-TX, tissues stained for GFP were incubated overnight at 4°C with 1:500 Alexa Fluor 488-conjugated donkey anti-rabbit, Alexa Fluor 594-conjugated donkey anti-mouse, and Alexa Fluor 649-conjugated donkey anti-rat (all from Invitrogen). For visualization of RFP, staining was with a 1:500 dilution of Alexa Fluor 594-conjugated donkey anti-rabbit and Alexa Fluor 649-conjugated donkey anti-rat. After exposure to secondaries, tissues were then washed in PBS-TX, mounted onto

**Table 5.** Split GAL4 lines used in study.

| Split line | Target cell | AD | DBD |
|---|---|---|---|
| **MB065b** | DAN-f1 (+DAN-c1) | TH-p65ADZp in attP40 | R72B05-ZpGdbd in attP2 |
| MB145 | DAN-f1 (+DAN-c1) | R15B01-p65ADZp in attP40 | R72B05-ZpGdbd in attP2 |
| MB 196C | DAN-i1 | R58E02-p65ADZp in attP40 | R36B06-ZpGdbd in attP2 |
| MB198B | DAN-k1 | R58E02-p65ADZp in attP40 | R71D01-ZpGdbd in attP2 |
| MB316B | DAN-j1 | R58E02-p65ADZp in attP40 | R93G08-ZpGdbd in attP2 |
| **MB328B** | DAN-d1 | R82C10-p65ADZp in attP40 | R32F01-ZpGdbd in attP2 |
| MB340C | DAN-j1 | R93D10-p65ADZp in attP40 | R12G04-ZpGdbd in attP2 |
| MB440B | DAN-h1 | R30G08-p65ADZp in attP40 | R17D06-ZpGdbd in attP2 |
| **MB586B** | DAN-c1 | TH-p65ADZp in attP40 | R72G06-ZpGdbd in attP2 |
| SS00616 | DAN-k1 | 71D01-p65ADZp in VK00027 | 17D06-ZpGdbd in attP2 |
| **SS00860** | MBON-j2 | w; R89G07-p65ADZ; MKRS/TM6B | R24E12-ZpGdbd in attP2 |
| **SS00867** | MBON-a1 | w; R93G12-p65ADZ; MKRS/TM6B | R52E12-ZpGdbd in attP2 |
| **SS01417** | MBON-a1 | w; R52E12-p65ADZp | R93G12-ZpGdbd in attP2 |
| SS01624 | MBIN-l1 | w; R84D07-p65ADZ | R37G09-ZpGdbd in attP2 |
| **SS01671** | APL | R21D02-p65ADZp | R55D08-ZpGdbd in attP2 |
| SS01696 | DAN-h1 | 76F05-p65ADZp in attP40 | 95H02-ZpGdbd in attP2 |
| SS01702 | DAN-c1 | VT054895-p65ADZ in attP40 | R53C05-ZpGdbd in attP2 |
| **SS01705** | MBON-d1 | R11E07-p65ADZp in attP40 | R52H01-ZpGdbd in attP2 |
| **SS01708** | MBON-b1,b2 | R12G03-p65ADZp in attP40 | 21D02-ZpGdbd in attP2 |
| **SS01716** | DAN-g1 | R14E06-p65ADZp in attP40 | R27G01-ZpGdbd in attP2 |
| **SS01725** | MBON-h1,h2 | R20A02-p65ADZp in attP40; MKRS/TM6B | R28A10-ZpGdbd in attP2 |
| SS01755 | DAN-g1 | R46F09-p65ADZp | R14E06-ZpGdbd in attP2 |
| **SS01757** | DAN-k1 | w; R48F09-p65ADZp; MKRS/ TM6B | R27A11-ZpGdbd in attP2 |
| SS01771 | MBON-i1 | w; 65A05-p65ADZ; MKRS/TM6B | 14C08-ZpGdbd in attP2 |
| **SS01949** | DAN-h1, -i1, -j1 | VT026700-p65ADZp in attP40 | VT058464-ZpGDBD in attP2 |
| SS01958 | OAN-e1 | VT023826-p65ADZp in attP40 | R75F01-ZpGdbd in attP2 |
| SS01959 | MBON-b1,b2 | VT027952-p65ADZp in attP40 | R26A02-ZpGdbd in attP2 |
| **SS01962** | MBON-k1 | VT033301-p65ADZp in attP40 | R27G01-ZpGdbd in attP2 |
| SS01972 | MBON-j1 | VT057469-p65ADZp in attP40 | 12C11-ZpGdbd/ TM3 in attP2 |
| **SS01973** | MBON-j1 | VT057469-p65ADZp in attP40 | R18D09-ZpGdbd in attP2 |
| **SS01980** | MBON-k1 | VT020613-p65ADZp in attP40 | VT033301-ZpGdbd in attP2 |
| **SS02006** | MBON-a2 | w; 93G12-p65ADZ; MKRS/TM6B | 71E06-ZpGdbd in attP2 |
| SS02121 | MBON-g1,g2 | R21D06-p65ADZp in attP40 | R23B09-ZpGdbd in attP2 |
| **SS02130** | MBON-g1,g2 | w; R23B09-p65ADZp; MKRS/ TM6B | R21D06-ZpGdbd in attP2 |
| **SS03066** | DAN-c1 | VT054895-p65ADZ in attP40 | VT057278-ZpGdbd in attP2 |
| **SS04112** | MBON-b1,b2 | VT027952-p65ADZp in attP40 | HAV5; CyO/Sco; 21D02-ZpGDBD in attP2 |
| **SS04231** | MBON-d2 | VT032899-p65ADZp in attP40 | HAV5; CyO/Sp; 87G02-ZpGDBD in attP2 |

*Table 5 continued on next page*

*Table 5 continued*

| Split line | Target cell | AD | DBD |
|---|---|---|---|
| **SS04268** | OAN-g1 (sVPMmx) | VT012639-p65ADZp in attP40 | VT016127-ZpGdbd in attP2 |
| **SS04328** | MBON-f2 | VT033301-p65ADZp in attP40 | VT029593-ZpGdbd in attP2 |
| **SS04484** | MBIN-l1 | R37G09-p65ADZp in attP40 | VT007174-ZpGdbd in attP2 |
| **SS21716** | MBIN-b1,b2 | VT048835-p65ADZp in attP40 | VT026664-ZpGdbd in attP2 |
| SS21789 | MBON-c1 | VT050247-p65ADZp in attP40 | VT050247-ZpGDBD in attP2 |
| **SS25844** | OAN-g1 (sVPMmx) | VT040569-p65ADZp in attP40 | VT061921-ZpGdbd in attP2 |
| SS36248 | MBON-f2 | VT016795-p65ADZ in attP40 | VT029593-ZpGdbd in attP2 |
| **SS36923** | OAN-e1 | VT054895-p65ADZ in attP40 | HAV5; CyO/Sco; 75F01-ZpGDBD in attP2 |
| SS04559 | MBON-e2 | w; 65A05-p65ADZ; MKRS/TM6B | VT045663-ZpGDBD in attP2 |

poly-lysine-coated coverslips, dehydrated through an ethanol series, cleared in xylenes, and mounted in DPX mountant (Sigma-Aldrich). Nervous systems were imaged on a Zeiss LSM 510 confocal microscope at ×40 with optical sections taken at 2 μm intervals. LSM files were contrast-enhanced as necessary and *z*-projected using ImageJ (http://rsbweb.nih.gov/ij/). Reagents are summarized in *Table 6*.

## Acknowledgements

We are grateful to Scarlett Pitts and Todd Laverty of the Janelia FlyCore for dealing with *Drosophila* maintenance and setting up the needed crosses. Members of the Janelia FlyLight team, including Geoffrey Meissner, Susana Tae, Jennifer Jeter, Scott Miller, and Sophia Protopapas, were involved in dissection, immunocytochemistry, and imaging. We thank Lynn Riddiford for critical comments on the manuscript. The research was funded by HHMI.

**Table 6.** Reagents used in this study.

| Reagent | Source | Catalog # |
|---|---|---|
| Mouse anti-bruchpilot | Developmental Studies Hybridoma Bank | Nc82-s |
| Rat anti-N cadherin | Developmental Studies Hybridoma Bank | DN-Ex #8 |
| Mouse anti-neuroglian | Developmental Studies Hybridoma Bank | BP 104 |
| Mouse anti-Fasciclin II | Developmental Studies Hybridoma Bank | 1D4 |
| Rabbit anti-DsRed | ClonTech | #632496 |
| Normal donkey serum | Jackson ImmunoResearch | #017-000-121 |
| AF488 donkey α-rabbit | Jackson ImmunoResearch | #711-545-152 |
| AF488 donkey α-mouse | Jackson ImmunoResearch | #711-585-151 |
| AF594 donkey α-rabbit | Jackson ImmunoResearch | #711-585-152 |
| AF594 donkey α-mouse | Jackson ImmunoResearch | #711-585-151 |
| AF649 donkey α-rat | Jackson ImmunoResearch | #711-605-153 |
| Mifepristone (RU-486) | Sigma-Aldrich | #M8046-100mg |
| S2 – Schneider's Insect Medium | Sigma-Aldrich | #S01416 |
| DPX mountant | Electron Microscopy Sciences | #13512 |

## Additional information

### Funding

| Funder | Grant reference number | Author |
|---|---|---|
| Howard Hughes Medical Institute | | James W Truman Tzumin Lee |

The funders had no role in study design, data collection and interpretation, or the decision to submit the work for publication.

### Author contributions

James W Truman, Conceptualization, Data curation, Formal analysis, Investigation, Visualization, Methodology, Writing – original draft, Project administration; Jacquelyn Price, Data curation, Supervision, Investigation, Methodology; Rosa L Miyares, Data curation, Formal analysis, Visualization, Methodology, Writing – review and editing; Tzumin Lee, Conceptualization, Resources, Formal analysis, Methodology, Writing – review and editing

### Author ORCIDs

James W Truman ⓘ http://orcid.org/0000-0002-9209-5435
Tzumin Lee ⓘ http://orcid.org/0000-0003-0569-0111

### Decision letter and Author response

Decision letter https://doi.org/10.7554/eLife.80594.sa1
Author response https://doi.org/10.7554/eLife.80594.sa2

## Additional files

### Supplementary files

• MDAR checklist

### Data availability

All data generated or analyses in this study are included in the manuscript and the supporting images.

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
