## [Editor Report]

The complete metamorphosis of higher insects is one of the most fascinating and complex processes in nature. The discrepancy in form and function between larvae, pupa, and adult insects is breathtaking, begging the question of how these forms and functions can so seamlessly follow each other. For the highest-order brain center of the insects, the mushroom body, the authors provide a masterpiece analysis of this process at the cellular level. Given the breadth and depth of the data that the authors present, the current study will serve as a reference for the field of developmental neuroscience for many years to come; indeed, the study data are eagerly awaited in the field.

---

## [Decision Letter]

**Decision letter after peer review:**

Thank you for submitting your article "Metamorphosis of memory circuits in *Drosophila* reveal a strategy for evolving a larval brain" for consideration by *eLife*. Your article has been reviewed by 3 peer reviewers, and the evaluation has been overseen by a Reviewing Editor and K VijayRaghavan as the Senior Editor. The following individuals involved in the review of your submission have agreed to reveal their identity: Astrid Deryckere (Reviewer #3).

Essential revisions:

As you will see from the individual reviews, this work is greatly appreciated as a tour de force and will be of lasting value to science and the community. Therefore, some more effort in communicating these important results to both novices and experts will be invaluable. Some key points, primarily from reviewer 3 are summarised below but please do also carefully read each of the individual reviews.

1) One major concern is that although the hypothesis that the larval brain is derived seems very plausible, it is confusing to read for example in the abstract that the larval brain is the *Drosophila*'s second brain, while during the development of the animal, this is the first structure to be seen. This can be explained more clearly. As stated by one reviewer ""to start the abstract and the introduction with the evolutionary ancient scenario of development in hemimetabolous insects, NOT with the *Drosophila* individual-development scenario. This is because a process is best explained from the start, ie from the hemimetabolous beginnings. It is then much easier to grasp that the *Drosophila* larval stage is a pretty forced 'detour' from the ancient trajectory."

2) In a similar way, the results of this paper do not explain how the larval brain has evolved, but instead shed new perspectives on the process of metamorphosis. Therefore, the manuscript would benefit from a change of title and re-writing.

3) The discussion specifically is very long and covers a lot of topics, not all central to the data presented in the paper. Please consider substantial shortening (potentially other parts can be used in a future review). Additionally, the section 'Relationship of cell fate to mushroom body compartments' is more appropriate in the results and might make the results easier to interpret.

*Reviewer #1 (Recommendations for the authors):*

In the abstract and introduction, I think a better job could be done to explain to general readers that the evolutionary and the individual trajectory in a sense are opposite. That is, the evolutionary trajectory is, in a sense, from an adult towards a larval brain, while the individual trajectory is from a larval to an adult brain. Most general readers, I think, would naively think both these trajectories are from larval ('worm-like') to adult, and I think it would be important to get this misunderstanding out of the way as early in the manuscript as possible.

I think the authors should try to knit in the MBINs into the scenario they present in the abstract, not only the MBONs.

It would seem better to me to start the abstract and the introduction with the evolutionary ancient scenario of development in hemimetabolous insects, NOT with the *Drosophila* individual-development scenario. This is because a process is best explained from the start, ie from the hemimetabolous beginnings. It is then much easier to grasp that the *Drosophila* larval stage is a pretty forced 'detour' from the ancient trajectory.

The figures are way too small, I think.

For the larval compartments, the authors use the established plain names and their abbreviations, e.g. shaft of medial lobe/ SHA, which is fine. Now, for the MBINs and MBONs names are established that read like eg DAN-h1, with the compartment indicated by the lower case letter, in this example h would stand for the SHA compartment. Again, this is fine.

However, in eg Figure 2B, C, and related figures, the sub-figures are labelled alphabetically (Figure 2B-a-b-c-d…), showing eg DAN-f1 in (d). I found this confusing. Maybe one could instead use Figure 2B-1-2-3-4…?

90: Also mention non-olfactory input (and introduce PNs as cell class).

97: Maybe better to mention already here up to which point do the KC axons prune back?

108: "the" input cells may not be properly understood by novices unless PNs and MBINs are clearly defined/ discriminated in the text.

110: Some MBINs are neither DANs nor OANs, right? Better define/ mention?

117: Maybe expressly say 10 compartments PLUS THE CALYX, as for novices it might not be clear that (and why) the calyx is not counted in.

135: Instead of "most" better give a percentage?

137: The authors nicely introduce the pairs of MBINs-MBONs… the most obvious question here is: for those MBINs and MBONs that do persist, do they stay together? As far as I understand the paper (p14, figure 10A, B), development seems to 'deliberately' make sure that they do not (excepting the calyx). This would be good to say here already (and maybe in the abstract). Given the rich functional data on the adult PPL1-01/MBON-11 pair, this may serve as an illustrative example. (To me, an important take-home message was that of all larval MBINs and MBONs, only the four medial lobe DANs die, while all others either remodel within the mushroom body or trans-differentiate into neurons with connections only outside the mushroom body. As a result, none of the larval MBIN-MBON pairs stays together (excepting the calyx))

Table 1 and Figure 2Ci: The established name is MBIN-c1, not DAN-c2, I think. Is there evidence that MBIN-c1 is dopaminergic? If so, how does this affect the assignment of the two adult PPL1-01s?

Table 1: Does "undetermined" and "unknown" mean the same? Better to use one expression only or define the difference.

141: How many are thus not-considered?

151: Easier to read if the sequence in the text matches the sequence in figure (from top to bottom).

Figure 4: Maybe the (A-I´) labels can go, and one could better put the larval names of the neurons to the left, and the adult names to the right. Similar for some of the other figures.

219: In relation to the adult APL, I am not so sure about the lobe-to-calyx feedback.

262: It is not so much about the homology itself, but about the interpretation of a character as ancestral or derived (in German 'Lesrichtung'), I think.

262: Maybe good to briefly explain here WHY this is difficult.

Figures: Maybe a good idea to use the icons for the larval and adult body throughout (as in Figure 1C).

Figure 6: This is a conceptually very important figure, I think. However, the point does not come across visually. You may consider:

in A) label as "Embryonic-born" and "Post-embryonic-born";

indicate that the preparations in A)left and A)right are from larvae and adults, respectively (?); you may use either the body-icons mentioned above, or lettering as "Embryonic-born @larval stage", or similar;

maybe add a sketch showing schematically how these neurons derive from the NB.

287 and following: Might be good to mention here that the post-embryonic-born MBINs seem to replace the dead larval DANs in the medial lobe; and that the post-embryonic-born MBONs supply the compartment which, as far as I understand from Figure 3B, would otherwise be 'orphaned' in adults.

Figures 8BC, 9, and 10C-F I found very difficult to get the point without the legend/ body text.

Figure 10B, legend: is the information regarding the colour code correct?

Line 414 and following: It might be interesting to note that both DAN-f1 (Weiglein et al. J Comp Neurol) and PPL1-01 (Aso and Rubin 2016 *eLife*, König et al. 2018 Learn Mem) are peculiar in that they support oppositely-valenced odour memories when activated before versus after presenting the odour (in adults this is also seen for PPL1-06 of the α-3 compartment).

DAN-c1 does not seem to be of either punishing or rewarding effect, by the way (Eschbach et al. 2020); this may be considered in Figure 10C.

422: Maybe there is a cut-past stump here?

409 and following: It is very generous of the authors that they apparently want to give the reader hints in which direction to look for the molecular genetics of trans-differentiation (likely based on unpublished data earmarked for a separate, later publication), and this should definitively remain. However, maybe this can be achieved in a shortened way.

638 and following: It might be wise to mention that the cases mentioned here are more imprinting-like, as compared to the more acute CS-US associations that the authors otherwise discuss.

670 and following: I think this description of 'the problem' and 'the solution' would be good to include in the abstract.

Contemplating the Results section and the authors' apparent love for their cells I was smiling when thinking "Ok, all these cells are 'exceptional'". After all, this might well reflect the biological reality that the larval neurons are all unique in structure, function, and/or developmental trajectory. And at the same time readers are unique – meaning the dataset is so rich that probably each and every reader has an idiosyncratic interest in knowing more about these neurons, or in having the data analysed in different ways. Below I list my points of interest, in the understanding that the authors are free to follow up on these or not, without justification for their decision being needed.

i) At least for those DANs that have individually been looked at in terms of their capacity to serve as reward or punishment, a separate table comparing that capacity in larvae to what was reported in adults would be useful.

ii) Thum and Gerber (2019) stressed that many of the MBINs and MBONs connect across hemispheres, forming what they called 'the mushroom body chiasm'. What is the fate of these interhemispheric connections across metamorphosis?

iii) Maybe a dedicated section about the calyx would be a good idea?

iv) 617 and Figure 10: In adults, there are some across-hemisphere, direct PPL1-01 to MBON-11 connections (Pavlowsky et al. 2018 Curr Biol) which are thought to enable a 'second-order conditioning'-like process within the compartment (König et al. 2019 Biol Lett) that would counteract the extinction process the authors mention, which operates across compartments.

*Reviewer #2 (Recommendations for the authors)*

I only have a few comments and questions that I think addressing will improve the script. I also noticed a few typos.

Abstract line 23 the authors use apostrophes instead of prime symbols. This happens through most of the text and figures too, although it is correct near the end of the script, line 684. It should be changed to prime throughout.

Line 72. 'while other(s) radically'.

Line 78. formatting error with references in square brackets.

Line 100. I think it would be worth mentioning here that there is further complexity to these main classes of KC. Eg γ dorsal, α-β posterior, etc.

Figures 2 and 3 show MBINs and MBONs, respectively. I think it would be good if the authors could somehow label those that are maintained vs those that remodel/transdifferentiate. Label for those that die is obvious!

Line 1060. Legend to Figure 3 'mushroom body output cells (MBONs)'. The N of course represents Neurons not Cells.

Line 324. DAN-di …'shows little change from larva to adult'. Would be good to name what it is in the adult. I think it's PPL103?

Line 327. Similar issue to above. To make it easier for the reader the authors should name the corresponding adult MBON to MBON-j1. I think it's MBON02?

Line 484-492. This section contains really important reference information comparing the larval to adult MB structures. I think this would be very helpful if it was positioned much earlier in the script, before analysing all the MBINs and MBONs.

Line 582-3. Long-term learning. Long-term memory. The references for this section are reviews rather than primary literature (eg. in adults Felsenberg 2017, 2018 for extinction/reconsolidation and Krashes 2009, Senapati 2019. I am not aware all of these phenomena have been demonstrated in larvae.

Line 600. PPL101 in hunger-state control of memory expression Krashes et al. 2009. MBON11 is Perisse et al. (2016). MBON11 was shown to inhibit other MBONs in Perisse et al. 2016 and not O(s)wald et al. 2015. The next use of Owald et al. 2015 is correct but it's Owald, not Oswald!

Line 603. dNFP typo of dNPF. No citation is provided for the discussion of this work. It should be Krashes et al. 2009.

Line 617. Again a review is cited for extinction and reconsolidation. Since the discussion focuses on the MBON12 junction this should be Felsenberg et al. 2017 and McCurdy et al. 2021.

Line 617-628. Is there a good citation for extinction in the larva?

Line 708. formatting error with references in square brackets.

*Reviewer #3 (Recommendations for the authors):*

I would like to start by congratulating the authors for their thorough mapping of the larval mushroom body input and output neurons throughout metamorphosis. As explained in the public review, my major concerns are the disconnection of the results with the introduction and discussion, as well as figures that are very hard to comprehend. I would suggest a major reorganization of the manuscript.

One major concern is that although the hypothesis that the larval brain is derived seems plausible, it is confusing to read for example in the abstract that the larval brain is the *Drosophila*'s second brain, while during the development of the animal, this is the first structure.

In a similar way, the results of this paper do not explain how the larval brain has evolved, but instead shed new perspectives on the process of metamorphosis.

Therefore, the manuscript would benefit from a change of title and major re-writing.

The discussion specifically is very long and covers a lot of topics, not all central to the data presented in the paper. Please consider substantial shortening (potentially in a future review). Additionally, the section 'Relationship of cell fate to mushroom body compartments' is more appropriate in the results and might make the results easier to interpret.

---

## [Author Response]

Essential revisions:1) One major concern is that although the hypothesis that the larval brain is derived seems very plausible, it is confusing to read for example in the abstract that the larval brain is the *Drosophila's* second brain, while during the development of the animal, this is the first structure to be seen. This can be explained more clearly. As stated by one reviewer ""to start the abstract and the introduction with the evolutionary ancient scenario of development in hemimetabolous insects, NOT with the Drosophila individual-development scenario. This is because a process is best explained from the start, ie from the hemimetabolous beginnings. It is then much easier to grasp that the Drosophila larval stage is a pretty forced 'detour' from the ancient trajectory."

We agree with this comment and have started the Introduction from the perspective of direct developing insects. We have also removed reference to a “second” brain.

2) In a similar way, the results of this paper do not explain how the larval brain has evolved, but instead shed new perspectives on the process of metamorphosis. Therefore, the manuscript would benefit from a change of title and re-writing.

We disagree with this point and would like to keep the title as is. As discussed at the end of the comments about point 3, we can provide some insight into how the larval brain has evolved.

3) The discussion specifically is very long and covers a lot of topics, not all central to the data presented in the paper. Please consider substantial shortening (potentially other parts can be used in a future review). Additionally, the section 'Relationship of cell fate to mushroom body compartments' is more appropriate in the results and might make the results easier to interpret.

The Discussion has been completely rewritten in line with the suggestions from the reviewers. Figures 8A, 9, 10C-F, and 11 of the old version have been removed. The discussion of neuronal remodeling strategies (Figure 8A) has been streamlined and instead of using four examples of changes of circuit function going from larva to adult (Figure 10C-F), we have reduced the discussion to only one example (New Figure 9C) which illustrates the complexities seen during metamorphosis. A proper comparison of the roles of these neurons in the larva and adult would be worth a paper in itself.

The line of the new Discussion is as follows: “Metamorphosis of mushroom body compartments”:

Figure 9A makes the point that 7 of the 10 larval compartments are incorporated into the adult MB (in that many of the same neurons function in both), while three larval compartments are not carried over into the adult. One has no adult counterpart while two are replaced with an adult-specific version (the adult vertical lobes)

“Transmitter stability and shifts through metamorphosis”: Figure 9B compares the transmitter output from the various adult compartments. Overall, there is a strong correspondence between the larva and adult systems but with a notable mismatch at the base of the lobes. As illustrated in Figure 9C is due to MBONs switching compartments rather than individual MBONs switching their transmitters.

“Circuit Level implications of changes through metamorphosis”: This section examines the circuit ramifications of compartment switching by MBONs/MBINs as well as trans-differentiation and addition of new neurons. Figure 10 confirms that changes in MBIN-MBON pairings occurs across the metamorphosing MB. At this level, there are no persisting connections that might support a memory trace through metamorphosis.

“Metamorphic changes of larval neurons” : Compared to the original version, this is a streamlined version of the range of changes that we see in neurons through metamorphosis, going from minor remodeling to trans-differentiation. It no longer has part of a figure associated with it (removal of old Figure 8A). One important point that we make is that one cannot use solely the extent of larva pruning to determine if a neuron is undergoing simple remodeling or is undergoing trans-differentiation.

“Relationship of lineage to metamorphic fates of MB neurons”: An important finding of the paper is the importance of trans-differentiation in recruiting neurons for the larval MB. The obvious question is how does this recruitment come about? Our finding that most recruits come from lineages that make permanent MBINs/MBONs is crucial because it allows us to focus on mechanisms that are known to control cellular phenotypes within a given lineage. The scenarios that we present in Figure 12B and C are admittedly speculative, but they are rooted in well understood mechanisms.

“The generation of the larval mushroom body”: This last section uses the lineage perspective explore some of the challenges that had to be overcome to evolve a larval mushroom body. Within the Holometabola, there has been selection to make a larva as rapidly as possible so that they can exploit temporary food sources (rotting fruit, dung, carcasses, etc.) before they are gone. Rapid embryogenesis, though, results in an early truncation of neurogenesis and relatively few neurons to make a larval brain. Based on the highly conserved mechanisms that determine neuronal phenotypes in insects, this truncation should result in the larva lacking some cell types that it needs to make a MB while having other cell types that it does not need. We find that trans-differentiation provides a mechanism to temporarily convert the latter into the former. We think that this is an important evolutionary insight.

Reviewer #1 (Recommendations for the authors):In the abstract and introduction, I think a better job could be done to explain to general readers that the evolutionary and the individual trajectory in a sense are opposite. That is, the evolutionary trajectory is, in a sense, from an adult towards a larval brain, while the individual trajectory is from a larval to an adult brain. Most general readers, I think, would naively think both these trajectories are from larval ('worm-like') to adult, and I think it would be important to get this misunderstanding out of the way as early in the manuscript as possible.I think the authors should try to knit in the MBINs into the scenario they present in the abstract, not only the MBONs.It would seem better to me to start the abstract and the introduction with the evolutionary ancient scenario of development in hemimetabolous insects, NOT with the *Drosophila* individual-development scenario. This is because a process is best explained from the start, ie from the hemimetabolous beginnings. It is then much easier to grasp that the Drosophila larval stage is a pretty forced 'detour' from the ancient trajectory.

This has been done in the revised introduction.

The figures are way too small, I think.

We have enlarged many of the figures.

For the larval compartments, the authors use the established plain names and their abbreviations, e.g. shaft of medial lobe/ SHA, which is fine. Now, for the MBINs and MBONs names are established that read like eg DAN-h1, with the compartment indicated by the lower case letter, in this example h would stand for the SHA compartment. Again, this is fine.However, in eg Figure 2B, C, and related figures, the sub-figures are labelled alphabetically (Figure 2B-a-b-c-d…), showing eg DAN-f1 in (d). I found this confusing. Maybe one could instead use Figure 2B-1-2-3-4…?

We apologize for the unnecessary complexity from the numbering of the subfigures in Figures2 and 3. We thought it would make it easier to go from the summary (part B) to the images (part C). In the revised figures (now Figures3 and 4). We no longer use subfigure lettering – e.g. Figure 2Ba, and the larval neurons are indicated by their name in the summary (part A) and the images (part B). To allow us to make the images larger, we have not shown the larval anatomy for the four neurons that die, or the four neurons whose fates are unknown.

90: Also mention non-olfactory input (and introduce PNs as cell class).

Done

97: Maybe better to mention already here up to which point do the KC axons prune back?

The compartments that lose γ axons during pruning are now indicated in Figure 2A.

108: "the" input cells may not be properly understood by novices unless PNs and MBINs are clearly defined/ discriminated in the text.

This should now be clear from the Introduction.

110: Some MBINs are neither DANs nor OANs, right? Better define/ mention?

This has been included in the legend for Figure 3.

117: Maybe expressly say 10 compartments PLUS THE CALYX, as for novices it might not be clear that (and why) the calyx is not counted in.

This has been done in the revision.

135: Instead of "most" better give a percentage?

This has been done.

137: The authors nicely introduce the pairs of MBINs-MBONs… the most obvious question here is: for those MBINs and MBONs that do persist, do they stay together? As far as I understand the paper (p14, figure 10A, B), development seems to 'deliberately' make sure that they do not (excepting the calyx). This would be good to say here already (and maybe in the abstract). Given the rich functional data on the adult PPL1-01/MBON-11 pair, this may serve as an illustrative example. (To me, an important take-home message was that of all larval MBINs and MBONs, only the four medial lobe DANs die, while all others either remodel within the mushroom body or trans-differentiate into neurons with connections only outside the mushroom body. As a result, none of the larval MBIN-MBON pairs stays together (excepting the calyx))

As suggested the PPL1 01/MBON-11pair is part of the specific illustration provided in Figure 9C. Figure 10A then goes farther in showing that none of the MBIN-MBON pairings (except the calyx) persist through metamorphosis.

Table 1 and Figure 2Ci: The established name is MBIN-c1, not DAN-c2, I think. Is there evidence that MBIN-c1 is dopaminergic? If so, how does this affect the assignment of the two adult PPL1-01s?

We have used MBIN-c1. DAN-c1 and MBIN-c1 are extremely similar except that MBIN-c1 appears to be born later in embryogenesis.

Table 1: Does "undetermined" and "unknown" mean the same? Better to use one expression only or define the difference.

We now use unknown throughout.

141: How many are thus not-considered?

Stated in the Introduction that we determined the fates of 80% of the MBINs/MBONs from Saumweber *et al.*

151: Easier to read if the sequence in the text matches the sequence in figure (from top to bottom).

Both figure and text have been revised.

Figure 4: Maybe the (A-I´) labels can go, and one could better put the larval names of the neurons to the left, and the adult names to the right. Similar for some of the other figures.

Now Figure 5 and associated Figure supplements. The A-I labels are gone and refer to cells only by their names. The same solution was made for the nest figure [old Figure 5; new Figure 6].

219: In relation to the adult APL, I am not so sure about the lobe-to-calyx feedback.

We have removed this statement.

262: It is not so much about the homology itself, but about the interpretation of a character as ancestral or derived (in German 'Lesrichtung'), I think.

In comparing neurons groups across species present the same challenge as comparing genes, in the context of gene duplication and subsequent diversification. This paper is probably not the place to tackle issues of “homologous” versus “orthologous” versus “paralogous”.

262: Maybe good to briefly explain here WHY this is difficult.

So done!

Figures: Maybe a good idea to use the icons for the larval and adult body throughout (as in Figure 1C).

Thanks for the idea!

Figure 6: This is a conceptually very important figure, I think. However, the point does not come across visually. You may consider:in A) label as "Embryonic-born" and "Post-embryonic-born";indicate that the preparations in A)left and A)right are from larvae and adults, respectively (?); you may use either the body-icons mentioned above, or lettering as "Embryonic-born @larval stage", or similar;maybe add a sketch showing schematically how these neurons derive from the NB.

This is now Figure 7. We have redrafted it to make it more comprehensible.

287 and following: Might be good to mention here that the post-embryonic-born MBINs seem to replace the dead larval DANs in the medial lobe; and that the post-embryonic-born MBONs supply the compartment which, as far as I understand from Figure 3B, would otherwise be 'orphaned' in adults.

This should now be obvious in the rewriting of the section.

Figures 8BC, 9, and 10C-F I found very difficult to get the point without the legend/ body text.

Figure 8BC has been redrafted to make the new Figure 12B,C. The first part of the Figure – part A diagrammatically summarizes the information used to establish cell identity within a lineage. The following panels B and C then uses that diagrammatic format. This should help the reader. Figure 9 has been removed and Figure 10C-F has been removed. Part of Figure 10C retained in the new Figure 9C – the latter includes an intermediate pupal condition to stress the retargeting of arbors through metamorphosis.

Figure 10B, legend: is the information regarding the colour code correct?

The color code is correct. There were some mistakes in part A but these have been corrected

Line 414 and following: It might be interesting to note that both DAN-f1 (Weiglein et al. J Comp Neurol) and PPL1-01 (Aso and Rubin 2016 eLife, König et al. 2018 Learn Mem) are peculiar in that they support oppositely-valenced odour memories when activated before versus after presenting the odour (in adults this is also seen for PPL1-06 of the α-3 compartment).DAN-c1 does not seem to be of either punishing or rewarding effect, by the way (Eschbach et al. 2020); this may be considered in Figure 10C.

These are points that are worth making and have been included.

422: Maybe there is a cut-past stump here?

Fixed

409 and following: It is very generous of the authors that they apparently want to give the reader hints in which direction to look for the molecular genetics of trans-differentiation (likely based on unpublished data earmarked for a separate, later publication), and this should definitively remain. However, maybe this can be achieved in a shortened way.

If not shortened, at least it should be easier to read!

638 and following: It might be wise to mention that the cases mentioned here are more imprinting-like, as compared to the more acute CS-US associations that the authors otherwise discuss.

The papers do deal with associative conditioning.

Reviewer #2 (Recommendations for the authors)I only have a few comments and questions that I think addressing will improve the script. I also noticed a few typos.Abstract line 23 the authors use apostrophes instead of prime symbols. This happens through most of the text and figures too, although it is correct near the end of the script, line 684. It should be changed to prime throughout.

This has been corrected.

Line 72. 'while other(s) radically'.

Changed in the revision.

Line 78. formatting error with references in square brackets.

brackets have been removed.

Line 100. I think it would be worth mentioning here that there is further complexity to these main classes of KC. Eg γ dorsal, α-β posterior, etc.

This has been done in the revison.

Figures 2 and 3 show MBINs and MBONs, respectively. I think it would be good if the authors could somehow label those that are maintained vs those that remodel/transdifferentiate. Label for those that die is obvious!

Now Figures3 and 4. The neurons that stay with the MB are indicated by the dashed arrows leading to their adult form. Those that trans-differentiate are labeled as “T”

Line 1060. Legend to Figure 3 'mushroom body output cells (MBONs)'. The N of course represents Neurons not Cells.

Corrected

Line 324. DAN-di …'shows little change from larva to adult'. Would be good to name what it is in the adult. I think it's PPL103?

As now noted at the start of the Discussion, in most places we designate the neurons using both names – i.e. DAN-d1/PPL1 03.

Line 327. Similar issue to above. To make it easier for the reader the authors should name the corresponding adult MBON to MBON-j1. I think it's MBON02?

See above

Line 484-492. This section contains really important reference information comparing the larval to adult MB structures. I think this would be very helpful if it was positioned much earlier in the script, before analysing all the MBINs and MBONs.

This is an excellent suggestion. The new Discussion now starts with the section “Metamorphosis of mushroom body compartments”. It provides

Line 582-3. Long-term learning. Long-term memory. The references for this section are reviews rather than primary literature (eg. in adults Felsenberg 2017, 2018 for extinction/reconsolidation and Krashes 2009, Senapati 2019. I am not aware all of these phenomena have been demonstrated in larvae.

In the revision, we have cited primary literature, rather than reviews.

Line 600. PPL101 in hunger-state control of memory expression Krashes et al. 2009. MBON11 is Perisse et al. (2016). MBON11 was shown to inhibit other MBONs in Perisse et al. 2016 and not O(s)wald et al. 2015. The next use of Owald et al. 2015 is correct but it's Owald, not Oswald!

We have omitted hunger state in the revision.

Line 603. dNFP typo of dNPF. No citation is provided for the discussion of this work. It should be Krashes et al. 2009.

This is no longer relevant

Line 617. Again a review is cited for extinction and reconsolidation. Since the discussion focuses on the MBON12 junction this should be Felsenberg et al. 2017 and McCurdy et al. 2021.

These have now been cited

Line 617-628. Is there a good citation for extinction in the larva?

Not relevant with the new revision

Line 708. formatting error with references in square brackets.

These have been corrected

Reviewer #3 (Recommendations for the authors):I would like to start by congratulating the authors for their thorough mapping of the larval mushroom body input and output neurons throughout metamorphosis. As explained in the public review, my major concerns are the disconnection of the results with the introduction and discussion, as well as figures that are very hard to comprehend. I would suggest a major reorganization of the manuscript.One major concern is that although the hypothesis that the larval brain is derived seems plausible, it is confusing to read for example in the abstract that the larval brain is the *Drosophila's* second brain, while during the development of the animal, this is the first structure.In a similar way, the results of this paper do not explain how the larval brain has evolved, but instead shed new perspectives on the process of metamorphosis.Therefore, the manuscript would benefit from a change of title and major re-writing.

As stated above, we think that the end of the Discussion does relate to how a larval brain evolves. The evolution of the larval brain is faced with constraints related to the shortened period of embryonic development and the highly conserved temporal and spatial mechanisms that insects use to generate their neuronal phenotypes. These constraints result in a potential mismatch between the neurons that are needed and those that are actually made (revealed by the adult phenotypes of these neurons). The larva then turns to trans-differentiation to temporarily transform unneeded (or dead) neurons into the missing cell types to build its larval circuits.

We think that these ideas provide some new insights into how a larval brain may have evolved and that our title is appropriate.